https://doi.org/10.1038/s42003-020-0975-4　**OPEN**

# Tau affects P53 function and cell fate during the DNA damage response

Martina Sola[1,2,5], Claudia Magrin[1,2,5], Giona Pedrioli [1,3], Sandra Pinton[1], Agnese Salvadè[1], Stéphanie Papin[1,6] & Paolo Paganetti [1,4,6✉]

Cells are constantly exposed to DNA damaging insults. To protect the organism, cells developed a complex molecular response coordinated by P53, the master regulator of DNA repair, cell division and cell fate. DNA damage accumulation and abnormal cell fate decision may represent a pathomechanism shared by aging-associated disorders such as cancer and neurodegeneration. Here, we examined this hypothesis in the context of tauopathies, a neurodegenerative disorder group characterized by Tau protein deposition. For this, the response to an acute DNA damage was studied in neuroblastoma cells with depleted Tau, as a model of loss-of-function. Under these conditions, altered P53 stability and activity result in reduced cell death and increased cell senescence. This newly discovered function of Tau involves abnormal modification of P53 and its E3 ubiquitin ligase MDM2. Considering the medical need with vast social implications caused by neurodegeneration and cancer, our study may reform our approach to disease-modifying therapies.

[1] Neurodegeneration Research Group, Laboratory for Biomedical Neurosciences, Ente Cantonale Ospedaliero, Torricella-Taverne, Switzerland. [2] Faculty of Biomedical Sciences, Members of the PhD in Neurosciences Program, Università della Svizzera Italiana, Lugano, Switzerland. [3] Member of the International PhD Program of the Biozentrum, University of Basel, Basel, Switzerland. [4] Faculty of Biomedical Sciences, Università della Svizzera italiana, CH-6900 Lugano, Switzerland. [5] These author contributed equally: Martina Sola, Claudia Magrin. [6] These authors jointly supervised this work: Stéphanie Papin, Paolo Paganetti. ✉email: paolo.paganetti@eoc.ch

Tauopathies are disorders of Tau protein deposition best represented by Alzheimer's disease (AD), where Tau accumulation in neurofibrillary tangles of the brain correlates with the clinical course in terms of number and distribution[1,2]. Also, mutations in the *MAPT* gene encoding for Tau lead to frontotemporal dementia with Parkinsonism 17[1,2]. Since Tau is a microtubule-associated protein, an accepted concept explaining the pathogenesis of tauopathies is that abnormal phosphorylation and folding cause Tau detachment from microtubules, Tau accumulation, and neuronal dysfunction[3,4]. In addition to microtubule association, Tau localizes in the cell nucleus and binds DNA[5–8] and also forms a complex together with P53, Pin1, and PARN regulating mRNA stability through polyadenylation[9]. Nuclear Tau was shown to have a role in DNA protection, whereby heat or oxidative stress cause nuclear Tau translocation[10]. Enhanced DNA damage was observed in Tau-KO neurons when compared to normal neurons[11]. We reported that drug-induced DNA damage also causes Tau nuclear translocation and affects Tau phosphorylation[12]. Notably, checkpoint kinases controlling DNA replication and cell cycle following a DNA damage phosphorylate Tau[13]. Together with chromosomal abnormalities found in AD-derived fibroblasts[14] and increased DNA damage in AD brains[15,16], the emerging function of Tau in DNA stability offers an alternative role of Tau in neurodegeneration and, importantly and insufficiently investigated, also in the DNA damage response (DDR). DNA is continuously damaged by genotoxic agents originating from the environment or generated intracellularly. The integrity of the genome is ensured by an efficient DDR signaling network regulating cell cycle and the DNA repair machinery, but also the activation of cell death or senescence when DNA damage persists. DDR deregulation causes accumulation of DNA errors and genomic instability, both implicated in age-related pathologies as cancer and neurodegenerative disorders[17].

In order to evaluate a role of Tau in this process, we depleted Tau in human cells and then carefully analyzed the DDR. We demonstrate that Tau deficiency renders cells less sensitive to DNA damage-induced apoptosis, which is counterbalanced by increased senescence. We show that this activity of Tau is mediated through a P53 modulation. Overall, our findings propose a role of P53 in tauopathies and a role of Tau in P53 dysregulation, a key event in oncogenesis.

## Results

**Generation and characterization of Tau-KO and Tau-KD cells.** We opted the use of human SH-SY5Y neuroblastoma cells for generating Tau knock-out (Tau-KO) cells by the CRISPR-Cas9 technology and Tau knock-down (Tau-KD) cells by shRNA interference (Fig.1). For disruption of the *MAPT* gene, we designed gRNAs targeting Cas9 endonuclease to two sequences in the first coding *MAPT* exon. CRISPR-Cas9 cell lines were screened for Tau expression by fluorescent confocal microscopy and immune protein blotting with the human-specific N-terminal Tau13 antibody. So, we identified cell lines devoid of Tau (Fig.1a and Supplementary Fig. 7a). Since the Tau13 epitope is within the Cas9-targeted exon, false negatives may perhaps result from in-frame indels or abnormal mRNA processing. With the HT7 antibody against amino acid 159–164 of Tau$_{441}$, we confirmed the isolation of Tau-KO lines lacking full-length or truncated Tau expression (Fig. 1a and Supplementary Fig. 7a). We finally selected the cell lines 232P and 231K presenting alleles modified at the expected gRNA-sites by indels causing frame-shifts into stop codons within the same exon (Fig.1a). The 231A cell line underwent an unsuccessful CRISPR-Cas9 procedure and had normal Tau expression (Fig. 1a).

To obtain Tau-KD cells, we screened shRNAs targeting the coding sequence or the 3′ untranslated region of the Tau mRNA. Culturing shRNA transduced cells in the presence of puromycin resulted in the isolation of cell populations with constitutive down-regulation of Tau for three shRNAs as shown by immune staining and western blot (Fig.1b and Supplementary Fig. 7b).

**Tau deficiency protects against DNA damage-induced apoptosis.** Persistent DNA damage induces cell death or senescence. Thus, as a functional readout for the DDR, we assessed the cytotoxicity following a mild exposure to etoposide[18], a DNA topoisomerase II inhibitor causing double-stranded DNA breaks (DSBs). Cell viability was first tested with the well-established LDH and the MTS assays. Tau-KO cells exposed to a short (30 min) 60 μM etoposide treatment did not release LDH in the culture medium and more efficiently converted MTS when compared to Tau-expressing cells, which exhibited substantial etoposide-dependent cytotoxicity in both assays (Fig. 2). To test the involvement of apoptosis, we immune-stained cells for cleaved active caspase-3 (clCasp3), an initiator of apoptosis. Whilst <1% of the untreated Tau-expressing cells were positive for clCasp3, etoposide exposure increased the apoptotic population to 13–15%, apoptosis was induced in only 4–5% of Tau-KO cells (Fig.2). The presence of activated clCasp3 in Tau-expressing cells exposed to etoposide and its almost complete absence in Tau-KO cells was confirmed by western blot analysis with the same antibody for the cleaved enzyme form (Fig. 2 and Supplementary Fig. 8). As a whole, we found a positive association between Tau expression and DSB-induced apoptosis in SH-SY5Y cells.

**Tau depletion induces cellular senescence.** In alternative to cell death, unresolved DNA damage may provoke cellular senescence[17]. Inhibition of cyclin-dependent kinase by p21 causes cell cycle arrest and induction of senescence[19,20]. When compared to untreated conditions, at three recovery days after etoposide exposure (Fig. 3a), higher amounts of p21 were detected by western blot in both Tau-expressing and Tau-KO cells (Fig. 3b). When comparing the extent of this effect in the absence or the presence of Tau, we found that Tau depletion increased p21 both at basal conditions as well as after etoposide treatment when compared to wt cells (Fig. 3b). The increased amount of p21 present in Tau-KO cells suggests that Tau-depletion may prone cells to enter a senescence state further accelerated in the presence of DSBs. We determined the number of cells entering in a senescent state based on their mean cell size and by the senescence-associated β-galactosidase (SA-βGal) staining procedure at mild acidic conditions. When compared to untreated conditions, significantly increased cell size and SA-βGal-positive cells were found at three recovery days after etoposide exposure both for wt and Tau-KO cells (Fig. 3b). Again, Tau-depleted cells at basal conditions displayed a larger proportion of senescent cells in terms of cell size, SA-βGal staining and p21 expression (Fig. 3b). A consistent observation was made also for Tau-KD cells when compared to control shRNA cells (Fig. 3c, d). We concluded that reduced expression of endogenous Tau changed the fate of SH-SY5Y cells as a consequence of DSBs, favoring cellular senescence induction at the expense of activation of programmed cell death.

**DNA damage and DDR activation are not reduced.** DSBs lead to rapid recruitment and phosphorylation of the H2A histone family member at the site of DNA damage, and so the presence of γH2A-X is utilized as a surrogate marker of DNA damage. We first performed an accurate etoposide dose-response by in-cell

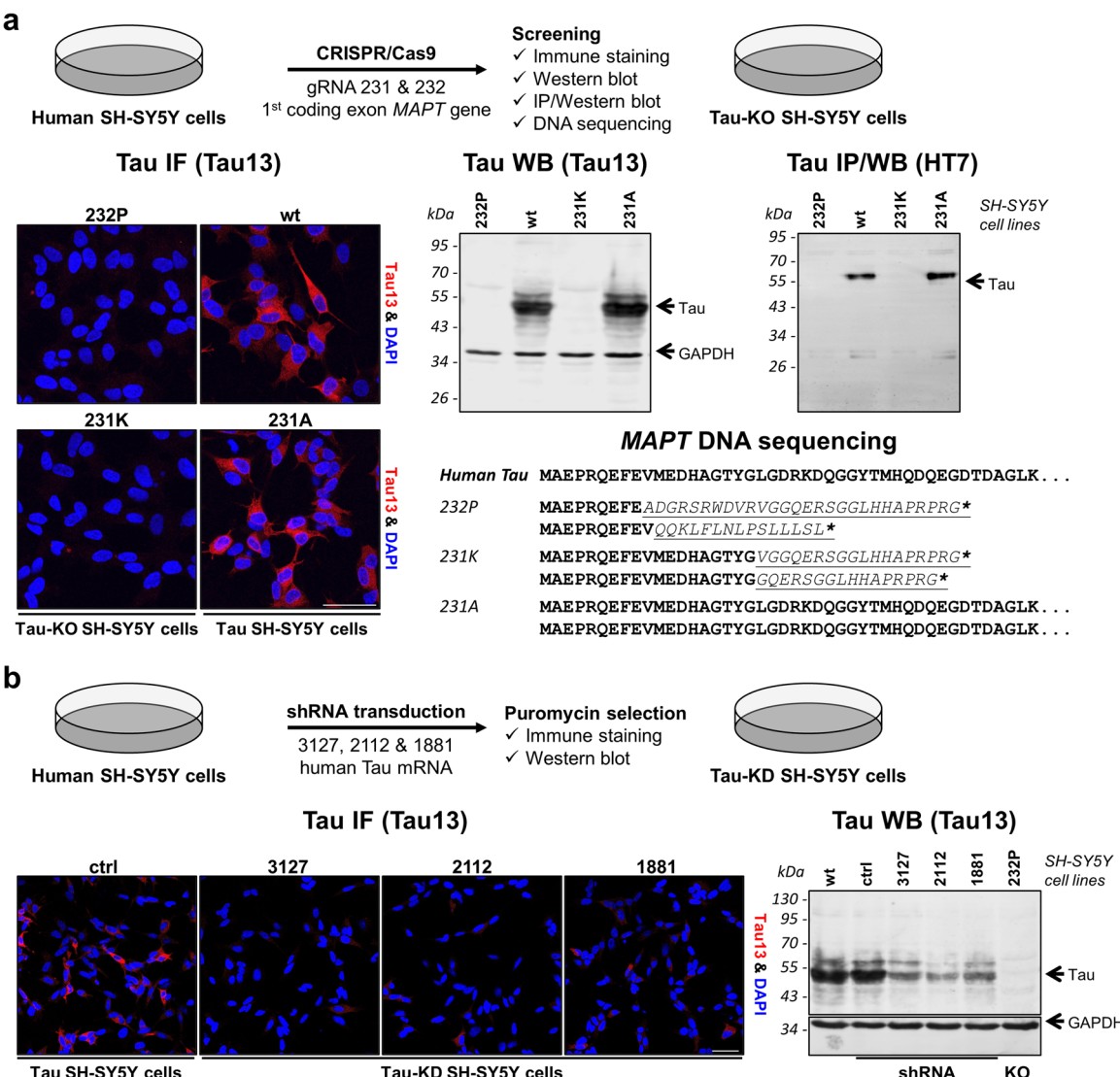

**Fig. 1 Generation of Tau-KO and Tau-KD SH-SY5Y cells. a** Scheme of the procedure used to generate CRISPR-Cas9-targeted cells and their characterization. Immune staining was performed with Tau13 antibody and nuclear staining with DAPI, western blot with Tau13 (loading control GAPDH) and immune precipitation and western blot with HT7 antibody, parental cells (wt) served as control. Amino acid sequences of the first *MAPT* coding exon in all lines demonstrate successful CRISPR-Cas9-editing causing frameshift (underlined in italics) into early stop codons (asterisks) for both alleles of 232P and 231K cells. **b** Scheme of procedure used to generate Tau-KD cell lines and their characterization by immune staining and western blot for Tau expression when compared to parental cells (wt) or cells transfected with the parental shRNA plasmid (ctrl). Scale bar 50 μm.

western for γH2A-X staining normalized by nuclear DAPI staining and observed an increase detection of γH2A-X total staining in Tau-KO cells when compared to control cells (Fig. 4a). The Comet assays is a direct measure of the magnitude of DSBs in single cultured cells. No difference between control and Tau-KO cells was found at the end of the etoposide treatment or during the recovery, which was rapid and complete before the 6 h washout time point independently on the presence or absence of Tau. However, Tau deletion led to more DSBs at basal conditions (Fig. 4b). Our data were thus consistent with a role of Tau in DNA-protection[10,11]. In contrast, the relatively small increase in etoposide-mediated DNA damage in Tau-KO cells inadequately explained reduced DNA damage-induced cell death in Tau-depleted cells. To corroborate this observation, we performed an etoposide dose-response and determined by confocal microscopy the presence of immune-stained nuclear γH2A-X and of the DSB-activated forms of ATM and Chk2[21,22]. This demonstrated a robust and dose-dependent induction of all three markers at

30 min after etoposide treatment (Fig. 4c). The difference between wt and Tau-KO cells was relatively minor and suggested a slightly stronger activation of the early DDR in Tau-KO cells, although the results obtained at 0 and 6 h recovery were less conclusive (Fig. 4d). Overall, the modest and somehow opposite effect of Tau depletion on the early DDR when compared to cell death induction, suggested a downstream contribution of Tau in modulating cell death.

**Tau modulates DDR-dependent stabilization of P53 protein**. A key DDR regulator is the tumor suppressor protein P53, which first halts cell division and then dictates cell fate when DNA damage persists[23,24]. To check the requirement of P53 for apoptosis induction in our cell model, we transduced cells with viral pseudoparticles and isolated stable P53 shRNA expressing cells (Supplementary Fig. 1a). The effect of the shRNA was negligible at basal conditions, i.e. when the cells maintain a minimal amount of P53 due to its efficient degradation. In

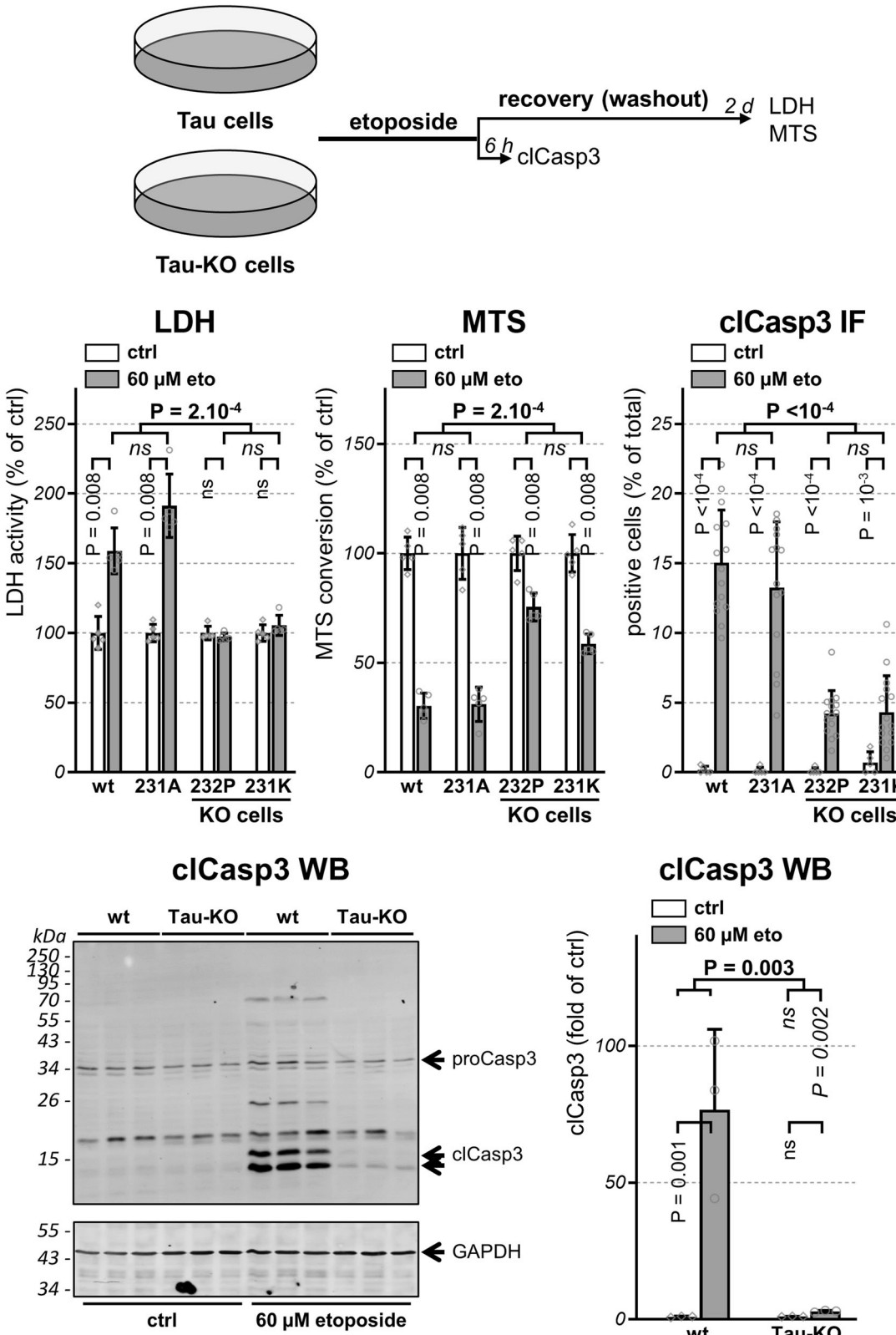

**Fig. 2 Tau deficiency confers resistance to etoposide-induced apoptosis.** Scheme representing the design of the experiment with parental and 231A (Tau) cells compared to 232P and 231K (Tau-KO) cells treated 30 min with 60 μM etoposide and recovered as indicated before analysis. LDH and MTS values are shown as percentage of parental cells (wt), mean ± SD of five biological replicates. To measure activation of apoptosis, percent positive cells for cleaved-caspase-3 (clCasp3) is determined on confocal microscope images and normalized for total DAPI-positive cells, mean ± SD of five images for the untreated cells (ctrl) and of 15 images for etoposide-treated cells (60 μM eto), $n > 500$ cells/condition, representative experiment of $n > 3$ biological replicates. Activated clCasp3 was also analyzed by western blot with GAPDH as loading control and 15 and 17 kDa clCasp3 quantified by normalization with GAPDH, mean ± SD ($n = 3$ biological triplicates). Statistical analysis by independent measures ordinary two-way ANOVA, source of variation for cell lines (in bold), multiple Bonferroni pairwise comparisons for treatment between lines (in italics) or for each line (in vertical).

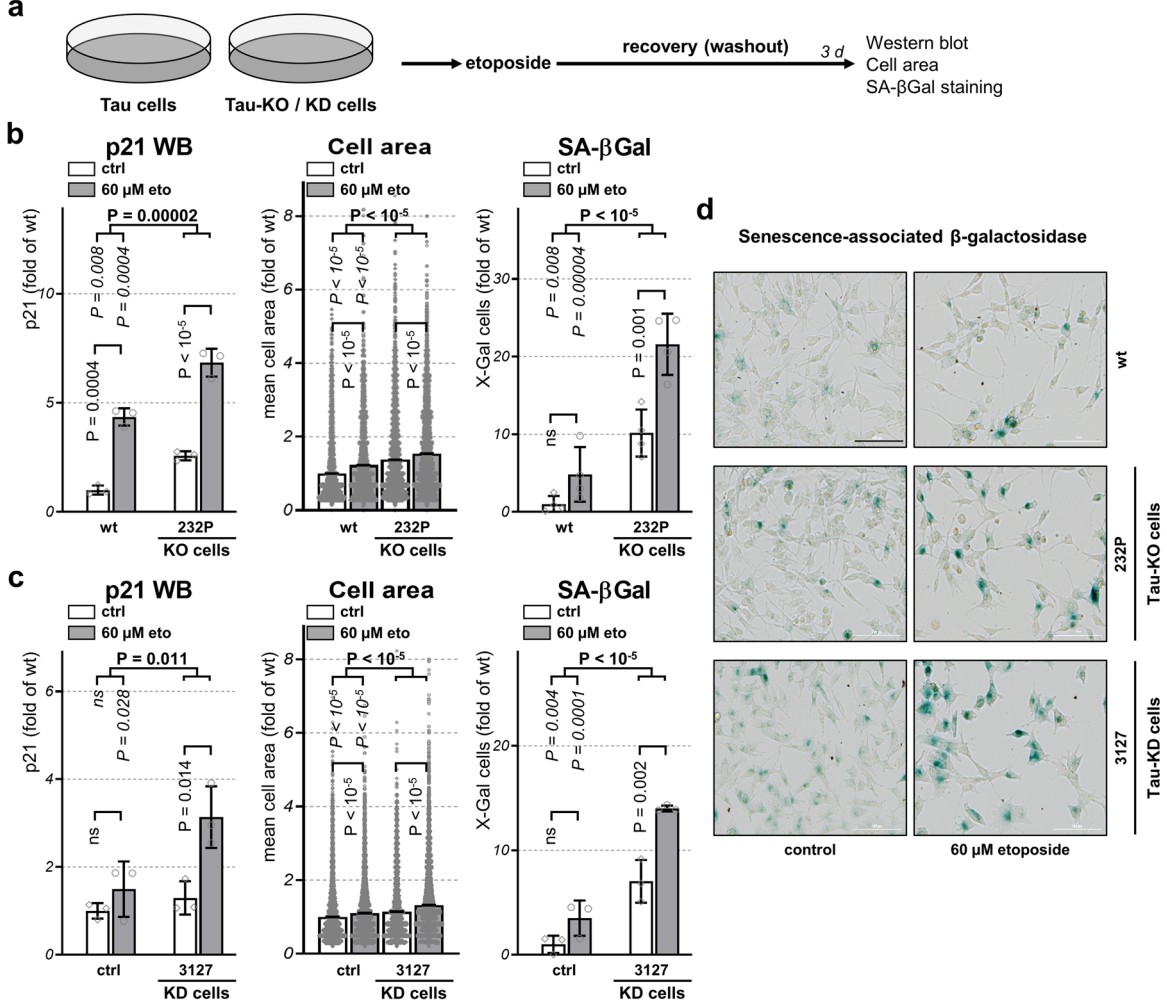

**Fig. 3 Tau depletion increases cellular senescence. a** Scheme of the procedure followed to assess cellular senescence upon 30 min treatment with 60 μM etoposide followed by 3 days of recovery. **b** Quantification of p21 amount in cell lysates by western blot in parental (wt) or 232P (Tau-KO) cells under control conditions (ctrl) or following etoposide treatment (60 μM eto) normalized for GAPDH, mean ± SD of three biological replicates. Quantification of mean cell area and percent positive cells for senescence-associated β-galactosidase (SA- βGal) determined with a high-content microscope scanner, mean ± sem of four (Tau-KO cells) or three (Tau-KD) independent experiments, $n > 8000$ cells. Data are shown as fold of wt cells at basal conditions. **c** Same as in **b** for mock shRNA (ctrl) or Tau 3127 shRNA (Tau-KD) cells. **d** Representative images of SA-βGal staining (in blue), bright-field, scale bar = 100 μm. Statistical analysis by independent measures ordinary two-way ANOVA, source of variation for cell lines (in bold), multiple Bonferroni pairwise comparisons for treatment between lines (in italics) or for each line (in vertical).

contrast, P53-KD cells displayed reduced etoposide-dependent P53 stabilization when compared to control cells as demonstrated by western blot analysis with the monoclonal antibodies DO-1 and Pab 1801 and confirmed by immune staining with DO-1 (Supplementary Fig. 1b, c). Cell lysates obtained from the neuroblastoma cell line SK-N-AS carrying a homozygous deletion in the *TP53* gene and therefore not expressing P53[25], were used as a negative control for P53 immune detection. Etoposide treatment induced apoptosis in ~2% P53-KD cells compared to ~14% of the control cells (Supplementary Fig. 1d). These data confirmed the involvement of P53 in DSB-induced apoptosis also in SH-SY5Y cells.

Having exposed the contribution of P53 and Tau in modulating DNA damage-dependent apoptosis in SH-SY5Y cells, we next asked whether Tau may modulate P53 activation. Tau-KO cells presented reduced DNA damage-induced nuclear P53 when compared to Tau-expressing cells as shown by immune staining and western blot (Fig. 5a and Supplementary Fig. 2a, b). Reduced P53 was observed when Tau-KO cells were exposed to

30, 60 or 90 μM etoposide and let recover for 30 min or 6 h (Fig. 5a). Reduced etoposide-induced apoptosis in Tau-KO cells displayed a similar dose-dependent effect (Fig. 5b).

Further documenting the role of Tau in etoposide-induced cytotoxicity, re-expressing high levels of human Tau441 in Tau-KO cells (Supplementary Figs. 3a and 11) increased P53 stabilization in etoposide-treated cells (Supplementary Fig. 3b) and restored sensitivity in the LDH and clCasp3 assays (Supplementary Fig. 3c). In order to obtain reconstituted Tau expression at a level similar to that of endogenous Tau, in a second set of experiments Tau-KO cells were transiently transfected with a 1:10 mixture of Tau410 and GFP plasmids or of empty and GFP plasmids. Tau expression was then analyzed in GFP-positive cells co-transfected either with the Tau410 or the empty plasmid by immune staining. This led to determine a level of ectopic expression corresponding to ~2-fold that of endogenous Tau determined in parental SH-SY5Y cells (Supplementary Fig. 3d). Under these conditions, 6 h after etoposide exposure Tau410-transfected Tau-KO cells displayed increased P53 stabilization

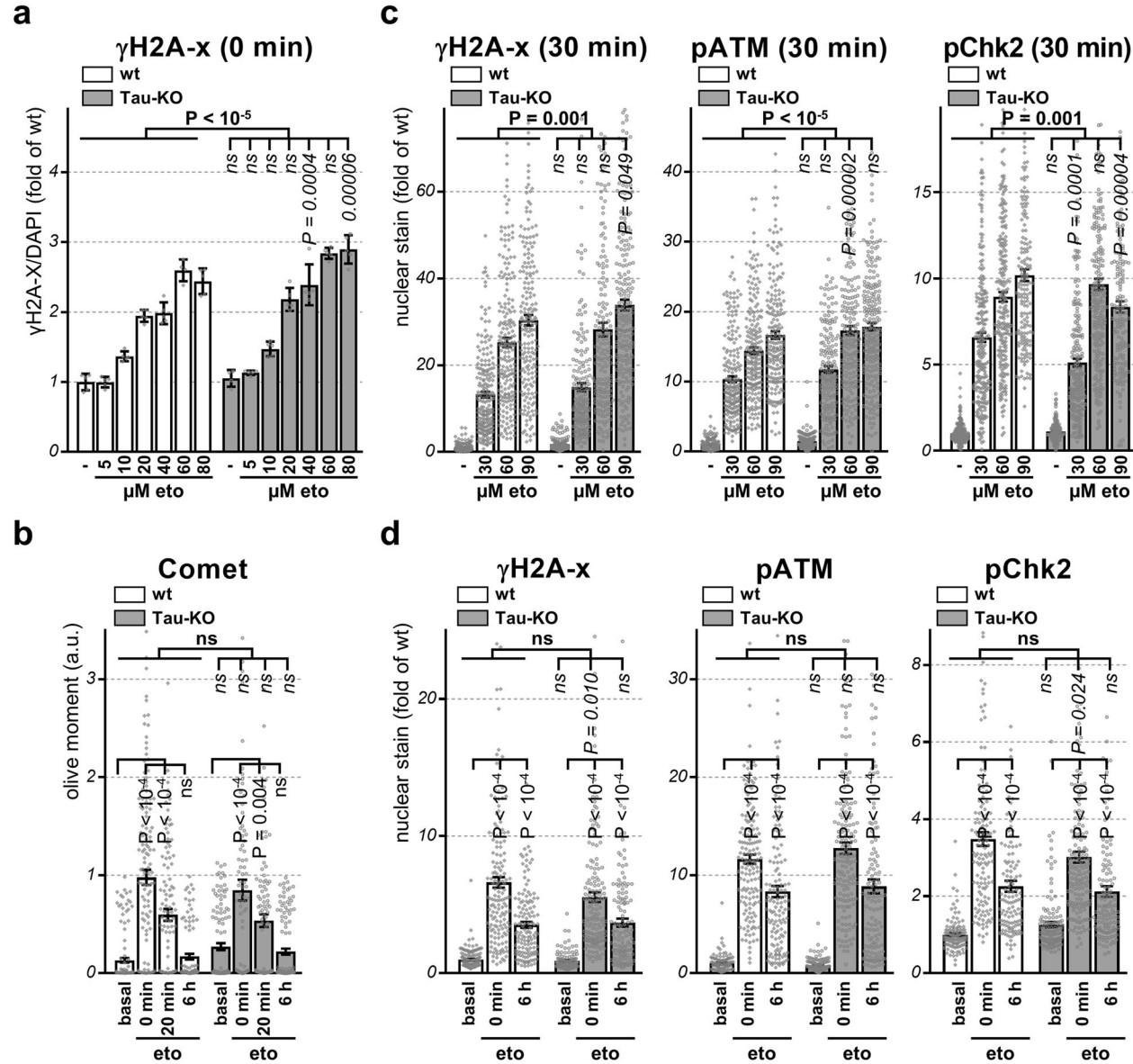

**Fig. 4 Reduced P53 in Tau-KO cells is not caused by DDR activation.** For all panels, parental (wt) or 232P (Tau-KO) cells were treated 30 min with the indicated etoposide concentrations and recovery times. **a** Mean intensity ± SD of γH2A-X staining normalized for DAPI staining by in-cell western is shown as fold of wt cells at basal conditions, $n = 5$ of biological replicates in a 96-well plate. Non-parametric independent Mann–Whitney U test between lines (in bold), or for each dose between lines (in italics). **b** Olive moment in the Comet assay is shown as mean ± sem, $n = 84$–$146$ cells/condition. **c, d** Mean intensity ± sem of single-cell nuclear γH2A-X, pATM or pChk2 staining (DAPI mask, ImageJ) is shown as fold of wt cells at basal conditions, $n > 100$ cells/condition distributed over five images. Statistical analysis by independent measures ordinary two-way ANOVA, source of variation for cell lines (in bold), multiple Bonferroni pairwise comparisons of each condition between lines (in italics) and of time points for each line (**b, d**, in vertical).

when compared to that detected in empty plasmid-transfected Tau-KO cells (Supplementary Fig. 3e).

Tau-KD-cells with reduced Tau-expression corresponding to $71 \pm 1\%$ for the 3127 shRNA and $64 \pm 2\%$ for the 2112 shRNA (Fig. 5c) when exposed to etoposide also showed reduced P53 activation (Fig. 5d) and apoptosis (Fig. 5e) in a Tau-dose-dependent manner. On the other hand, $60 \pm 1\%$ reduced Tau in 1881 shRNA cells did not affect P53 protein level and apoptosis (Fig. 5c–e). Single-cell analysis of the whole Tau-KO or Tau-KD cell population revealed that when we applied a threshold just above background to count P53-positive cells, etoposide-dependent P53 stabilization was better described by a change in the relative number of P53-positive cells rather than by a gradual correlation between Tau and P53 expression (Fig. 5f).

**Reduced P53 and apoptosis occurs in other neuroblastomas.** In order to validate the observation made in SH-SY5Y cells, we tested the effect of Tau down-regulation in IMR5 and IMR32 human neuroblastoma cell lines. Similar to SH-SY5Y cells, these two cell lines express a wild-type functional P53[26,27]. Several other neuroblastoma cell lines were disregarded because P53 mutations were causing either constitutive activation or expression loss of P53[27]. Tau expression in IMR5 cells was down-regulated ~4-fold in the presence of the 2112 shRNA (Supplementary Fig. 4a). Under these conditions, we observed lower etoposide-induced P53 stabilization in Tau-KD when compared to mock-transduced IMR5 cells as determined by western blot and immune staining analysis (Supplementary Fig. 4b). Similar to what observed in SH-SY5Y cells, etoposide-induced clCasp3 was

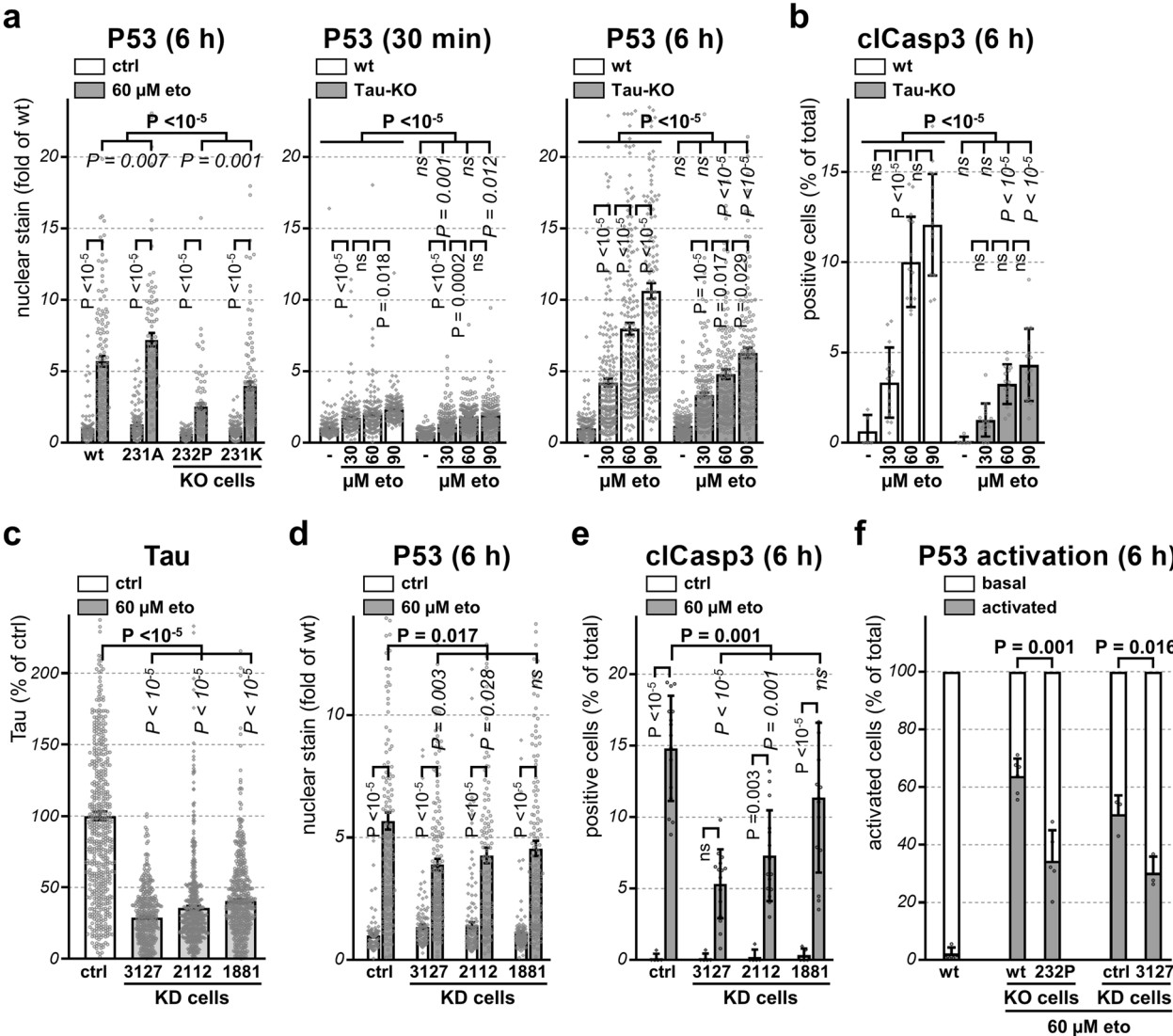

**Fig. 5 Tau depletion decreases P53 level and apoptosis.** For all panels the indicated cell lines (Tau-KO are 232P cells) were treated 30 min with the indicated etoposide concentrations and recovery times. **a** Mean intensity ± sem of single-cell nuclear P53 staining (DAPI mask, ImageJ) is shown as fold of wt cells at basal conditions, $n > 100$ cells/condition distributed over five images. **b** Percent clCasp3-positive cells are shown as mean ± SD of five images for the untreated cells and of 15 images for etoposide-treated cells, $n > 500$ cells/condition. **c** Mean intensity ± sem of single-cell Tau staining (tubulin mask, ImageJ) for the indicated cell lines is shown as percent of mock shRNA cells (ctrl), $n > 160$ cells/condition distributed over five images. **d** Mean intensity of single-cell nuclear P53 staining quantified as in **a**, $n > 380$ cell/condition distributed over 15 images from $n = 3$ biological replicates. **e** Percent clCasp3-positive cells quantified as in **b**, $n > 500$ cell/condition. **f** An arbitrary threshold was applied in order to count P53-positive cells as percentage ± SD of total DAPI-positive cell number, $n > 100$ cells/conditions. For the comparison between the four cell lines (**a**), non-parametric independent Mann–Whitney U test for genotype (in bold) and Kruskal–Wallis pairwise comparison of treatment for cell lines with same genotype (in italics) or for each line (in vertical). For the dose-dependency (**a**, **b**), non-parametric independent Mann–Whitney U test between cell lines (in bold), Kruskal–Wallis pairwise comparison for each dose (in italics) and between doses (in vertical). Non-parametric independent Mann–Whitney U test (**c**–**e**) between control and the three Tau-KD lines (in bold) and Kruskal–Wallis pairwise comparison for each Tau-KD line (in italic) and for each treatment (in vertical). Unpaired two-tailed t test with Welch's correction (**f**).

also reduced in Tau-depleted IMR5 cells (Supplementary Fig. 4c). The presence of the 3127 shRNA in IMR32 cells lowered Tau expression by ~40%, which resulted in reduced P53 stabilization and caspase-3 activation in cells exposed to the etoposide treatment (Supplementary Fig. 4d–f).

**Tau regulates P53 expression post-translationally.** To assess whether lower P53 protein level observed in Tau-KO cells was occurring by transcriptional or post-translational mechanism, we first determined by quantitative RT-PCR the amount of the P53 mRNA in wt and Tau-KO cells before or 6 h after the acute etoposide treatment. At basal conditions Tau-KO and Tau-KD cells showed a significant but modest increase in *TP53* transcription when compared to Tau-expressing cells. Etoposide exposure slightly increased the P53 transcript in all cell lines, but there was no difference when comparing treated Tau-expressing and treated Tau-KO cells (Fig. 6a, b). Overall, these data essentially dismissed the premise that the effect of Tau depletion on P53 stabilization occurred at the transcriptional level, rather suggesting a translational or post-translational control. On the other hand, etoposide treatment resulted in the expected

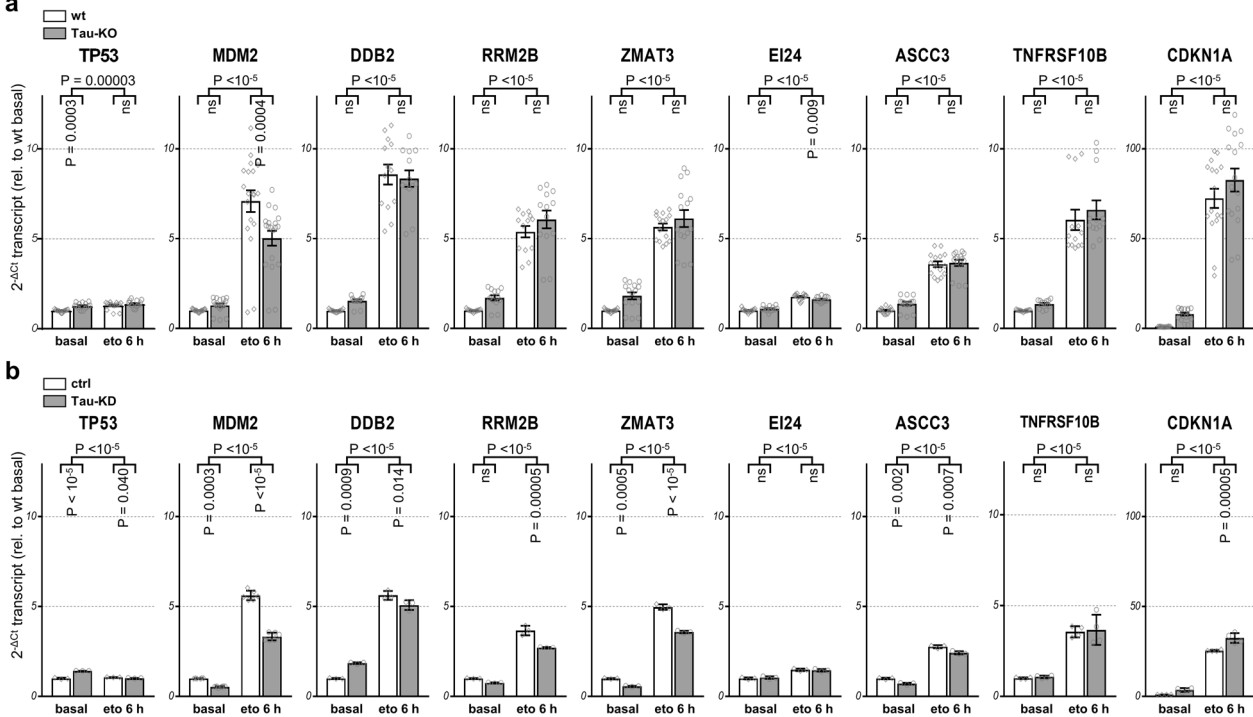

**Fig. 6 Differential regulation of P53 transcription targets.** Extracted RNA from parental (wt) and 232P (Tau-KO) cells in **a** or from control shRNA plasmid (ctrl) and 3127 (Tau-KD) cells in **b** at basal conditions or after 30 min 60 μM etoposide and 6 h recovery, was subjected to reverse-transcription and qPCR with primers specific for the indicated transcripts. Mean ± SD of relative mRNA levels ($n = 3$) shown as fold of the respective basal conditions for parental or control cells. Statistical analysis by independent measures ordinary two-way ANOVA, source of variation for cell lines (horizontal), multiple Sidak pairwise comparisons for treatment for each line (in vertical).

P53-dependent upregulation of *HDM2* transcription, but this was markedly reduced in Tau-KO and in Tau-KD cells (Fig. 6a, b), a result that was confirmed also at the MDM2 protein level (Supplementary Fig. 5). Analysis of additional direct targets of P53[28,29] showed a differential transcriptional response to etoposide in Tau-depleted cells. Whilst, transcription of the *EI24* gene was reduced in etoposide-treated Tau-KO cells, that of *RRM2B*, *TNFRSF10B*, *DDB2*, *ZMAT3*, *ASCC3* and *CDKN1A* was not affected in Tau-KO cells (Fig. 6a). Dysregulation of transcription of the P53 targets was more evident in Tau-KD cells, which showed reduced etoposide-induction for the *MDM2*, *DDB2*, *RRM2B*, *ZMAT3*, *ASCC3* transcripts, and no effect on the *EI24* and *TNFRSF10B* transcripts (Fig. 6b). Interestingly, as observed for the *CDKN1A* protein product p21 (Fig. 3c), also the *CDKN1A* transcript was increased in Tau-KD cells after etoposide treatment (Fig. 6b), whereas this did not reach significance in Tau-KO cells (Fig. 6a). A different regulation of direct P53-dependent genes after etoposide exposure, conveyed by a positive or negative difference in the degree of transcription activation between cells with normal or reduced Tau expression, substantiated a Tau-dependent modulation of P53 function at a post-translational level. Also, the heterogeneous response observed among the different P53 targets cannot be explained solely by a change in P53 protein stability but implied a more complex modulation of P53 activity.

**Tau affects P53 and MDM2 modification.** A post-translational clearance mechanism keeps P53 protein at low levels in the absence of a cellular stress[30]. This occurs mainly, but not exclusively, by the activity of the E3 ubiquitin ligase MDM2 (also known as HDM2) that associates with P53 to favor its degradation and interfere with its function[30,31]. We determined the

amount of nuclear MDM2 and found that induction of MDM2 was lower in Tau-KO cells when compared to Tau-expressing cells, possibly explained by reduced gene transcription (see above, Fig. 6), whereas MDM2 expression in untreated cells was not modulated by Tau (Supplementary Fig. 5). Post-translational modification of P53 through the action of DDR transducing kinases causes the dissociation of the P53-MDM2 complex and induces stress-dependent P53 stabilization. In the presence of DSBs, this occurs mainly by N-terminal phosphorylation of the P53 transcription-activation domain by the ATM-Chk2 axis[22]. We tested two small molecules interfering with this process. KU-55933 is an ATM inhibitor blocking ATM-dependent P53 phosphorylation thus preserving the P53-MDM2 complex and its degradation. Nutlin-3 binds to the P53-binding pocket of MDM2 thus inhibiting their association and degradation. KU-55933 had no effect on P53 and MDM2 expression when tested alone (Fig. 7a). As expected, adding the drug after etoposide treatment severely impaired DSB-induced P53 and MDM2 stabilization and also blocked apoptosis activation (Fig. 7a, b). Consistent with its mode of action, the presence of nutlin-3 led to a strong increase in P53 and MDM2, which was higher to that caused by etoposide but, notably, did not induce apoptosis (Fig. 7a, b). Nutlin-3 potentiated the effect of etoposide in terms of P53-MDM2 stabilization in wt cells. In Tau-KO cells exposed to etoposide, nutlin-3 eliminated the drop in MDM2 and partly also that in P53 (Fig. 7a, b). Determination of P53 phosphorylation at $S_{15}$ when normalized for total P53 protein showed a similar etoposide-dependent relative occupancy in Tau-KO cells when compared to Tau-expressing cells (Supplementary Fig. 6a). This was an unexpected result because amino-terminal P53 phosphorylation should stabilize P53 by interfering with the binding to MDM2, and thus increased P53 destabilization in Tau-KO cells should be reflected by a reduction in P53 phosphorylation. A

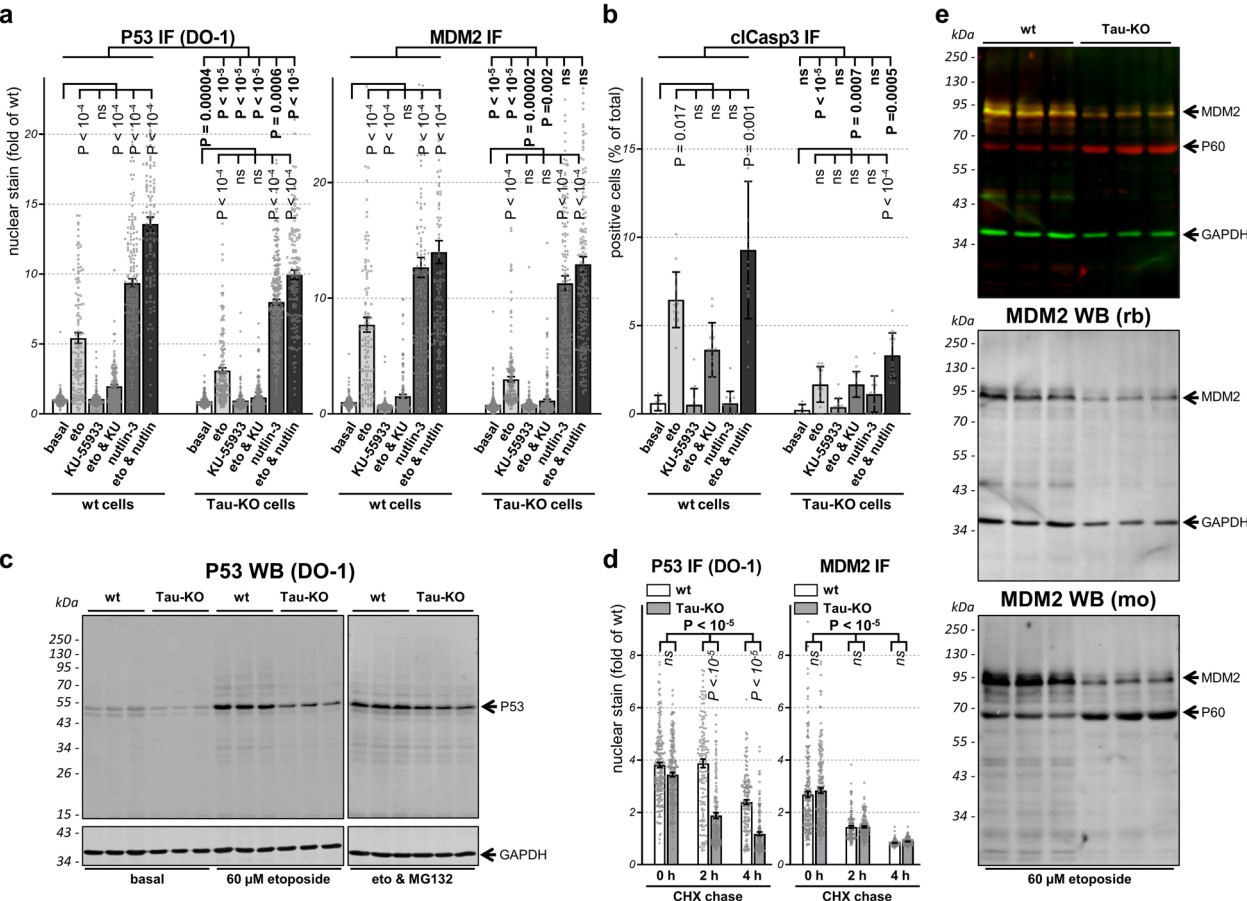

**Fig. 7 Role of P53 and MDM2 modifications for P53 function and stability. a**, **b** Parental (wt) or 232P (Tau-KO) cells treated 30 min without (basal) or with 60 μM etoposide and recovered for 6 h in the absence (eto) or presence of 10 μg/mL KU-55933 and/or 5 μg/mL nutlin-3. **a** Mean intensity ± sem of single-cell nuclear P53 or MDM2 (DAPI mask, ImageJ) shown as fold of basal conditions, n > 100 cells/condition distributed over five images. **b** Percent clCasp3-positive cells shown as mean ± SD of five images (basal) or 15 images (treatments), n > 500 cells/condition. Non-parametric independent samples test and Kruskal–Wallis pairwise comparison between cell lines (in bold) or for treatment for each cell line (in vertical). **c** Western bot analysis of P53 in parental (wt) or 232 P (Tau-KO) cells at basal conditions, after 30 min 60 μM etoposide and 4 h recovery without or with 10 μM MG132. GAPDH served as loading control. **d** Parental (wt) or 232P (Tau-KO) cells pre-treated for 30 min with 60 μM etoposide followed by 4 h with 10 μM MG132, were incubated with 25 μM of cycloheximide (CHX) for the indicated chase times. Single-cell nuclear P53 or nuclear MDM2 (DAPI mask, ImageJ) shown as fold of wt cells at basal conditions. Mean intensity ± sem of n > 100 cells/condition distributed over five images. Independent measures ordinary two-way ANOVA, source of variation for cell lines (bold), multiple Bonferroni pairwise comparisons of treatment for each line (in italic). **e** Parental (wt) or (Tau-KO) 232P cells treated for 30 min with 60 μM etoposide and 6 h recovery analyzed with a 90 kDa MDM2 rabbit antibody (green and middle panel with GAPDH as loading control) or a 60 and 90 kDa MDM2 mouse antibody (red and bottom panel).

possible explanation is that reduced DSB-dependent stabilization of P53 caused by the absence of Tau might be compensated by a change in P53 phosphorylation. Etoposide-induced P53 phosphorylation at $S_{15}$ was severely impaired by the ATM inhibitor KU-55933 (Supplementary Fig. 6b), but this only partially blocked apoptosis induction in wt cells and had no effect in Tau-KO cells when compared to etoposide-exposure alone (Fig. 7b). In addition, P53 stabilization by nutlin-3 did not involve $S_{15}$ phosphorylation (Supplementary Fig. 6b) and poorly induced apoptosis (Fig. 7b). Our attempts to analyze $pS_{46}$-P53 phosphorylation was unsuccessful as no signal was detected also under conditions of prolonged etoposide treatment both in Tau-expressing and Tau-KO cells. We concluded that in SH-SY5Y cells, DSB-induced apoptosis was at least in part dependent on P53 modification.

**Tau-depletion increases P53 degradation rate.** To address if Tau-KO cells displayed faster P53 degradation possibly accounting for the lower detection of P53 protein, we first

inhibited the ubiquitin-proteasome system by treating the cells with MG132. The presence of MG132 during the recovery phase from etoposide exposure restored P53 stabilization in Tau-KO cells but had no effect in wt cells, suggesting that the absence of Tau favored P53 degradation (Fig. 7c and Supplementary Fig. 9c).

Taking advantage of the fact that MG132 was able to restore similar P53 protein levels in wt and Tau-KO cells exposed to etoposide, we then analyzed the rate of degradation of P53 and MDM2 by removing MG132 and adding the translation inhibitor cycloheximide. Under these conditions we observed a faster P53 degradation rate at 2 and 4 h wash-out in Tau-KO cells when compared to wt cells (Fig. 7d). In contrast, no difference was observed in terms of MDM2 degradation (Fig. 7d).

Western blot analysis of MDM2 with the same rabbit antibody used for immune staining of the cells confirmed reduced MDM2 expression in Tau-KO cells exposed to etoposide when compared to wt cells (Fig.7e and Supplementary Fig. 9e). Interestingly, when using a mouse antibody for MDM2, we also detected a 60 kDa MDM2 form (Fig.7e and Supplementary Fig. 9e), likely representing the amino-terminal caspase-2 cleavage product of

## IP (FL393) / WB

**Fig. 8 Tau does not directly interact with P53.** Cell lysates of SH-SY5Y treated with 10 μM of MG132 to stabilize P53 expression, without (ctrl) or with (eto) a 30 min pre-treatment with 60 μM etoposide, were subjected to immune precipitation of endogenous P53 with a rabbit antibody (P53) or with a rabbit GFP antibody as negative IP control (GFP). Western blot analysis for co-precipitation of MDM2 or Tau with the respective mouse antibodies as indicated. The blots on the top show the analysis of the starting material (cell lysates), those on the bottom the immunoprecipitation (IP). The P53 blots are entirely shown, whereas for MDM2 and Tau, the blots were cut between the 55 kDa and the 95 kDa protein size markers and analyzed separately.

full-length 90 kDa MDM2[32,33]. An opposite effect of etoposide-exposure was obtained for these two forms of MDM2 in Tau-KO cells when compared to wt cells. Although reduced 90 kDa MDM2 detection was confirmed in Tau-KO cells, in the same cells we found that etoposide markedly increased 60 kDa MDM2 (Fig.7e and Supplementary Fig. 9e).

**Tau does not interact with P53**. In the presence of the proteasome inhibitor, the interaction between P53 and MDM2 was confirmed by a co-precipitation experiment both in the presence and in the absence of etoposide treatment (Fig. 8 and Supplementary Fig. 10). However, although etoposide increased the amount of MDM2 detected in cell lysates when compared to the control, for both conditions a similar amount of MDM2 was co-precipitated by P53; a result consistent with decreased P53-MDM2 interaction as a consequence of DNA damage. In contrast, we did not detect any interaction between P53 and Tau in the presence or absence of DNA damage (Fig. 8 and Supplementary Fig. 10), suggesting that the modulatory function of Tau on P53 stabilization may not occur by a direct interaction between the two proteins in SH-SY5Y cells.

The data obtained implied that in neuroblastoma cells Tau modulated P53 in a manner that went beyond a simple regulation of its stabilization, but accounted also for a deregulation of P53 and MDM2 post-translational modification ultimately affecting the activity and function of P53 in cell fate decisions dependent on DNA damage.

## Discussion

We report a new function of Tau as a regulator of DSB-induced cell fate and describe that this occurred by a deregulation of P53 activity. Our data support a role for Tau as a P53 modifier in neurodegeneration. Uncontrolled P53 activity in the presence of P53 or MDM2 mutations is one of the main pathomechanism of cancer[34]. Not surprisingly, the P53-MDM2 axis includes a variety of factors regulating their modification and localization[35]. Based on our data, Tau should now be listed as a modifier of wild-type P53 function, with possible implications in cancer biology.

The involvement of apoptosis in neurodegeneration is documented[36], but whether this is modulated by Tau remains questionable[37,38]. We show now that a brief DNA damaging insult positively associates Tau to programmed cell death and negatively to cellular senescence. Positive association to cell loss was already shown in Bloom's syndrome cells under continuous DNA damage for three days, although a distinction between cell death and senescence was not clear[39]. Age-dependent increase in senescent cells promotes tissue deterioration and neuronal dysfunction[40,41]. Moreover, increased senescent glia cells were found in a tauopathy model characterized by reduced soluble Tau and, remarkably, senescent cell removal prevented functional neuronal decline[42]. Nevertheless, direct evidence that Tau loss-of-function may promote senescence was not yet reported.

In the adult human brain, post-mitotic neurons express Tau in multiple alternative spliced isoforms that differ depending on the presence of up to two N-terminal inserts (0N, 1N, 2N) and on the presence of three or four microtubule-binding repeats (3R, 4R). In early development, 3R-Tau isoforms are predominant whereas in the adult brain the 3R and 4R isoforms are detected at a similar level, although they ratio is substantially altered in a peculiar manner for distinct tauopathies[43]. The cell lines used in our study (SH-SY5Y, IMR-32 and its subclone IMR-5) express predominantly the embryonic isoform 3R-Tau when actively dividing[44–46]. Our results showing that the effect of endogenous Tau deletion was reversed by ectopic expression of either 4R-Tau$_{441}$ or 3R-Tau$_{410}$ suggest that the modulatory role of Tau on P53 is possibly shared by all Tau isoforms. Tau phosphorylation in undifferentiated SH-SY5Y is increased[46], so at this time we cannot exclude that the modulatory effect of Tau on P53 may require modification of Tau by phosphorylation.

The balance between apoptosis and senescence is regulated by an intricate mechanism, which varies in response to distinct stressors[47]. In terms of DNA damage, crucial determinants are the nature and intensity of the stress. Since P53 drives both the induction of apoptosis and senescence, cell fate is finely tuned by changes in P53 kinetics and transcriptional activity under post-translational modification control[48]. The inference that Tau modifies P53 protein expression and the balance between cell death and senescence, suggests Tau as a modifier of P53 post-translation modification by acting on P53 modulators. P53 is modified by most types of substituents, which dictate the complex response to a wide range of cellular conditions[49]. Ubiquitination and phosphorylation control P53 stability, subcellular localization and transcriptional versus non-transcriptional activity, as well as cellular senescence[50] and apoptosis[51]. P53 acetylation may govern transcriptional-regulation of gene targets involved in growth arrest, and the choice to enter senescence or apoptosis[52]. In our cellular model, treatment with nutlin-3 restored P53 stabilization without involving P53 phosphorylation in Tau-depleted cells exposed to etoposide, but poorly reversed reduced activation of the apoptotic pathway. Nutlin-3 restored also the expression of the P53 negative regulator MDM2, which targets P53 for degradation, regulates its nuclear localization, and the interaction of P53 with transcriptional co-factors[35]. Therefore, considering only the protein amount of P53 or MDM2 inadequately explains the role of Tau as an effector balancing apoptosis and senescence. Additional P53-interacting protein such as WW domain-containing oxidoreductase (WWOX), which modulate Tau phosphorylation, may be involved in Tau-dependent regulation of P53[53,54].

**Table 1 Argeted gRNA sequences for human *MAPT* gene.**

| Oligo | Sequence |
| --- | --- |
| 231 | CACGCTGGGACGTACGGGTTGGG |
| 232 | CCGCCAGGAGTTCGAAGTGATGGG |

Tau has been described recently to be part of a complex containing P53, PARN and Pin1 involved in the regulation of mRNA stability through regulation of polyadenylation[9]. In this report, in HCT-116 colon cells Tau expression modulated the level of the P53 transcript and P53 that of Tau. In our cellular model, we also observed that Tau was able to modulate P53 expression, but we excluded that this occurred at the level of gene transcription and instead showed a post-translational mechanism. Nevertheless, the action of Tau on P53 degradation, on P53 activity as transcription factor and on P53 function as cell fate mediator reported herein for SH-SY5Y cells may well be regulated by a Tau complex similar to that described for HCT-116 cells. However, HCT-116 and other colon cancer cell lines modify Tau to an hyperphosphorylated form resembling the one deposited in tauopathy brains[55], suggesting a difference in function between pathogenic and physiological Tau.

Increased DNA damage has been reported as a consequence of Tau deletion and heat stress, indicating a protective role of Tau against DNA damage, which may require nuclear Tau translocation[10,11,35]. We reported that also etoposide exposure increased nuclear Tau and reduced its phosphorylation[12]. In the present study, determination of DNA integrity assessed by the Comet and γH2A-X assays confirmed the protective role of Tau, but rebuffed the possibility that decreased DNA damage in the absence of Tau may be the reason for decreased DSB-induced cell death. Moreover, we did not observe an overt effect on the ATM-Chk2 axis, implying that a downstream pathway was affected in Tau-KO cells.

P53 stabilization and apoptosis induction depended on the activity of the ATM-Chk2 axis, because they were stopped by the ATM inhibitor KU-55933. An alternative reading is that Tau may intervene on P53 modification or on its functional modulators. Whether this occurs in the cytosol, possibly based on its activity on microtubules, or following its nuclear translocation is yet to be examined. P53 has been shown to associate to and be regulated by microtubules[56,57]. Therefore, Tau may affect P53 interaction with the cytoskeleton by modulating microtubule dynamics. However, Tau is also present in the nucleus in normal and stress conditions and some functions associated to nuclear Tau were suggested[10,11,58–60].

The implication of the neurodegeneration-associated Tau protein in the biology of P53, the "guardian of the genome", is a thrilling finding that may explain the role of P53 and DDR dysfunction in neurodegeneration and the link between Tau and cancer. Abnormal P53 species are potential biomarkers of AD[61–63], the most common tauopathy with an high incidence of P53 mutations[64] and P53 deregulation[65]. Genetic manipulation of P53 family members in mice affects aging, cognitive decline, and Tau phosphorylation[66,67]. Cell cycle activators are upregulated in postmitotic neurons by stress conditions and tauopathies, possibly representing a cause for neurodegeneration[68,69]. Increased DNA damage is found in AD[15,16] and persistent DDR causes neuronal senescence and upregulation of pro-inflammatory factors[70]. Our finding that cellular senescence is increased by DSBs in Tau-KO cells is consistent with these observations.

Intriguingly, hyperphosphorylated and insoluble Tau is detected in some prostate cancer[71], FTDP-17 *MAPT* mutations

increase the incidence of cancer[72], and higher levels of phosphoSer199/202-Tau have value as predictors of non-metastatic colon cancer[73]. More recently several reports described that high Tau expression improves survival in several types of cancers[74–77]. Intriguingly, Tau deficiency resulting in reduced P53 stabilization reported herein provides a mechanism to explain why reduced Tau represents a negative prognostic marker. Moreover, since our data show that Tau expression also modulates etoposide cytotoxicity, we would like to propose that Tau protein level may acquire value as a response marker of genotoxic therapy.

Cancer and neurodegenerative diseases may involve common signaling pathways balancing cell survival and death[78–80] and may be defined as diseases of inappropriate cell-cycle control as a consequence of accumulating DNA damage. Epidemiological studies show an inverse correlation between cancer and neurodegeneration[79], although not consistently[81,82], and chemotherapy is associated to a lower predisposition for AD[83,84]. The study of Tau as a modifier of P53 and, importantly, P53 control of cell death and senescence is crucial because of the implication that Tau may modulate cell death and senescence in neurodegenerative tauopathies and in cancer. Considering the unmet medical need with vast social implications caused by these—unfortunately frequent —disorders, our finding holds sizeable scientific importance and may lead to innovative approaches for disease-modifying therapeutic interventions.

## Methods

**Cell culture and DNA transfections.** Human neuroblastoma IMR5, IMR32, and SK-N-AS were kindly provided by Dr. Chiara Brignole and Dr. Mirco Ponzoni from the IRCCS Istituto Giannina Gaslini in Genova. These cells and the human neuroblastoma SH-SY5Y cells (94030304, Sigma-Aldrich) were cultured in complete DMEM: Dulbecco's Modified Eagle Medium (61965–059, Gibco) supplemented with 1% non-essential amino acids (11140035, Gibco), 1% penicillin-streptomycin (15140122, Gibco) and 10% fetal bovine serum (10270106, Gibco). Cells were grown at 37 °C with saturated humidity and 5% $CO_2$, and maintained in culture for <1 month. Cells grown on poly-D-lysine (P6407, Sigma-Aldrich) were transfected with jetPRIME (114–15, Polyplus) or Lipofectamine 3000 (L3000008, Invitrogen) according to the manufacturer's instructions or with the calcium phosphate method[85].

**Targeted disruption of Tau expression.** For disruption of the *MAPT* gene encoding for Tau by the CRISPR-Cas9 method, the two gRNAs 231 and 232 (Table 1) targeted exon 2 containing the initiating ATG (ENST00000344290.9). Cells in six-well plates were transfected with the plasmid kindly provided by Dr. Zhang[86] (52961, Addgene) driving expression of one of the two gRNA, Cas9 nuclease and puromycin resistance. One day post-transfection, cells were transferred to 10 cm plates and incubated for 2 days with 20 μg/mL puromycin (P8833, Sigma-Aldrich). Single colonies were isolated, amplified, and stored in liquid nitrogen. The cDNA encoding for human Tau isoform of 441 amino acids (Tau441) in the expression plasmid pcDNA3 and selection in 0.5 mg/mL Geneticin (11811031, Gibco) served to generate reconstituted Tau expression in the Tau-KO 232P cell line.

**Sequencing of the targeted *MAPT* gene.** Cell pellets were resuspended in 400 μL TNES buffer (0.6% SDS, 400 mM NaCl, 100 mM EDTA, 10 mM Tris pH 7.5) and 0.2 mg/mL proteinase-K (Abcam, ab64220) under continuous shaking 3–4 h at 50 °C, and then supplemented with 105 μL of 6 M NaCl. Genomic DNA was precipitated with one volume of ice-cold 100% ethanol, washed with 100% ethanol and with 70% ethanol, and air-dried. The genomic fragment containing the CRISPR-Cas9-targeted regions was amplified by PCR with primers containing BamHI or XhoI restriction sites (Table 2) with the AccuPrime™ Pfx SuperMix (12344–040, Invitrogen). PCR reactions were purified with the GeneJET PCR purification kit (K0701, ThermoFisher Scientific) and subcloned in pcDNA3. DNA from single bacterial colonies were analyzed by restriction mapping with Ndel and XhoI or Van91I, in order to verify the presence and the *MAPT* origin of the inserts. Inserts with different size were selected in order to increase the chance of sequencing both alleles (Microsynth).

**Down-regulation of Tau or P53 expression.** Short hairpin RNAs (shRNAs, Table 3) were inserted in the pGreenPuro vector (SI505A-1, System Biosciences). The design of shRNAs targeting Tau or P53 was done following the manufacturer's instructions or with the tool provided by Dr. Hannon at http://katahdin.cshl.org//siRNA/RNAi.cgi?type=shRNA. Pseudo-lentiviral particles were produced in

**Table 2 PCR Primers (all specific for homo sapiens mRNAs).**

| Gene | Forward primer (5′–3′) | Reverse primer (5′–3′) |
|------|------------------------|-------------------------|
| MAPT | GATCAGGATCCGTGAACTTTGAACCAGGATGGC | GATCAGGATCCGTGAACTTTGAACCAGGATGGC |
| MDM2 | TGTTTGGCGTGCCAAGCTTCTC | CACAGATGTACCTGAGTCCGATG |
| TP53 | CCTCAGCATCTTATCCGAGTGG | TGGATGGTGGTACAGTCAGAGC |
| CDKN1A | AGGTGGACCTGGAGACTCTCAG | TCCTCTTGGAGAAGATCAGCCG |
| DDB2 | CCAGTTTTACGCCTCCTCAATGG | GGCTACTAGCAGACACATCCAG |
| ZMAT3 | GCTCTGTGATGCCTCCTTCAGT | TTGACCCAGCTCTGAGGATTCC |
| RRM2B | ACTTCATCTCTCACATCTTAGCCT | AAACAGCGAGCCTCTGGAACCT |
| ASCC3 | GATGGAAGCATCCATTCAGCCTA | CCACCAAGGTTCTCCTACTGTC |
| EI24 | GCAAGTAGTGTCTTGGCACAGAG | CAGAACACTCCACCATTCCAAGC |
| GAPDH | TGCACCACCAACTGCTTAGC | GGCATGGGACTGTGGTCATGAG |
| HPRT1 | TGACACTGGCAAAACAATGCA | GGTCCTTTTCACCAGCAAGCT |

**Table 3 Oligonucleotide annealed for shRNA sequences.**

| P53 | Sense | 5′gatccgactccagtggtaatctaccttcctgtcagagtagattaccactggagtctttttg 3′ |
|-----|-------|-----|
| | Antisense | 3′cgactccagtggtaatctaccttcctgtcagagtagattaccactggagtctttttgaatt 5′ |
| Tau 1881 | Sense | 5′gatcctggtgaacctccaaaatcacttcctgtcagatgatttggaggttcaccatttttg 3′ |
| | Antisense | 3′ctggtgaacctccaaaatcacttcctgtcagatgatttggaggttcaccattttttgaatt 5′ |
| Tau 2112 | Sense | 5′gatccaactgagaacctgaagcaccagcttcctgtcagactggtgcttcaggttctcagtgtttttg 3′ |
| | Antisense | 3′caactgagaacctgaagcaccagcttcctgtcagactggtgcttcaggttctcagtgtttttgaatt 5′ |
| Tau 3127 | Sense | 5′gatccagcagacgatgtcaaccttgtgcttcctgtcagacacaaggttgacatcgtctgccttttg 3′ |
| | Antisense | 3′cagcagacgatgtcaaccttgtgcttcctgtcagacacaaggttgacatcgtctgccttttgaatt 5′ |

HEK-293 cells (HEK 293TN, System Biosciences) with the pPACKH1 kit (LV500A-1, System Biosciences) by the calcium phosphate transfection method[85]. Particles were harvested 48–72 h later, concentrated on a 30K MWCO Macrosep Advance Spin Filter (MAP030C37, Pall Corporation), aliquoted and stored at −80 °C until use. Particle titers were determined at 72 h post-transduction by calculating the percent of GFP-positive cells and the mean GFP intensity. Tau-KD and P53-KD cells were obtained upon transduction and selection in 5 μg/mL puromycin for one to 2 weeks.

**Drug treatments**. Etoposide (100 mM stock in DMSO; ab120227, Abcam) treatment was followed by three washes with complete DMEM and cells were allowed to recover for the indicated times. The concentration of etoposide was adapted depending on the cell line, most treatments of SH-SY5Y cells were performed at 60 or 100 μM etoposide, whereas 15 μM etoposide was used for IMR5 cells and 30 μM etoposide for IMR32 cells. When specified, recovery was done in the presence of 5 μg/mL nutlin-3 (5 mg/mL stock in DMSO; SC-45061, Santa Cruz), 10 μg/mL of KU-55933 (10 mg/mL stock in DMSO; SML1109, Sigma-Aldrich), 10 μM MG132 (10 mM stock in DMSO, M7449, Sigma-Aldrich) or 25 μM cycloheximide (10 mg/ml stock in $H_2O$, 01810, Sigma-Aldrich). Vehicle DMSO was added in the controls.

**Immune assays**. For immune staining, cells were grown on poly-D-lysine coated eight-well microscope slides (80826-IBI, Ibidi). Cells were fixed in paraformaldehyde and stained[12] with primary antibodies 1 μg/mL Tau13 (sc-21796, Santa Cruz), 0.5 μg/mL $pS_{129}$-H2A-X (sc-517348, Santa Cruz), 0.13 μg/mL $pS_{1981}$-ATM (#13050, Cell Signaling), 0.2 μg/mL $pT_{68}$-Chk2 (#2197, Cell Signaling), 0.4 μg/mL P53 DO-1 (sc-126, Santa Cruz,), 1 μg/mL $pS_{15}$-P53 (ab223868, Abcam), 0.2 μg/mL MDM2 (#86934, Cell Signaling), 0.05 μg/mL clAsp$_{175}$-Caspase3 (#9661, Cell Signaling), 0.5 μg/mL alpha-tubulin (ab18251, Abcam). Detection of endogenous Tau entailed the testing of a number of commercial antibodies so to find the human-specific Tau13 monoclonal antibody against an amino-terminal epitope as reagent providing the most reliable detection of endogenous Tau. Although determination of DSBs by γH2A-X is mostly performed by counting positive nuclear foci, we noticed that at the concentration of etoposide used here, single foci were poorly discernible, we nevertheless confirmed that mean intensity of nuclear γH2A-X staining correlated with foci counting and decided to apply the first method for our quantifications of DNA damage. Detection by fluorescent laser confocal microscopy (Nikon C2 microscope) was done with 2 μg/mL secondary antibodies anti-mouse IgG-Alexa594 or -Alexa 488 (A-11032, A-11001, ThermoFisher Scientific) or anti-rabbit IgG-Alexa594 or -Alexa488 (A-11037, A-11034, ThermoFisher Scientific). Nuclei were counterstained with 0.5 μg/mL DAPI (D9542, Sigma-Aldrich). Images were usually taken with a line by line scan with a sequence of excitations, i.e. 405 nm laser with 464/40–700/100 nm emission filter, 488 nm laser with 525/50 nm emission filter, and 561 nm laser with 561/LP nm emission filter. ImageJ was used for all image quantifications.

For biochemical analysis by western blot, cells plated in 6 well plates were directly lysed in 40 μL SDS PAGE sample buffer (1.5% SDS, 8.3% glycerol, 0.005% bromophenol blue, 1.6% β-mercaptoethanol and 62.5 mM Tris pH 6.8) and incubated 10 min at 100 °C. For immune precipitation, cells from 10 cm plates were rinsed with PBS and collected by scraping and low speed centrifugation. Cell lysates were prepared in 400 μL ice-cold RIPA buffer (R0278, Sigma-Aldrich), supplemented with protease and phosphatase inhibitor cocktails (S8820, 04906845001, Sigma-Aldrich) and treated with benzonase (707463, Novagen) 15 min at 37 °C.

Protein immune precipitation was performed by a batch procedure using Protein G-Sepharose® beads (101241, Invitrogen) overnight at 4 °C with 40 μL 30% slurry beads and 1 μg of HT7 antibody (MN1000, Invitrogen) or P53 antibody (FL-393, bs-8687R, Bioss Antibodies) antibody. The cell lysates for P53 immune precipitation were cleared by centrifugation at 20,000 g per 10 min. For immune blots[12], we used 0.2 μg/mL Tau13 (sc-21796, Santa Cruz), 0.18 μg/mL GAPDH (ab181602, Abcam), 0.4 μg/mL P53 DO-1 (sc-126, Santa Cruz), 4 μg/mL P53 Pab 1801 (sc-98, Santa Cruz), 0.5 μg/mL $pS_{15}$-P53 (ab223868, Abcam), 0.1 μg/mL rabbit D1V2Z MDM2 (#86934, Cell Signaling), 0.2 μg/mL mouse SMP14 MDM2 (sc965, Santa Cruz), 0.05 μg/mL clAsp$_{175}$-Caspase3 (#9661, Cell Signaling), or 0.2 μg/mL p21 (sc53870, Santa Cruz). Immune precipitated Tau protein was detected with 0.1 μg/mL biotinylated HT7 antibody (MN1000B, ThermoFisher Scientific) and 0.2 μg/mL streptavidin-IRDye (926–32230, Licor Biosciences). In the co-IP experiments, the antibodies used were 0.4 μg/mL P53 DO-1, 1 μg/mL Tau13 and 0.1 μg/mL D1V2Z MDM2.

For in cell western, cells plated on poly-D-lysine-coated 96-well microtiter plates were fixed with cold 4% paraformaldehyde in PBS 15 min at 4 °C, stained with 0.5 μg/mL $pS_{129}$-H2A.X (sc-517348, Santa Cruz), anti-mouse IgG-IRDye680 (926–68070, Licor), and analyzed on a dual fluorescent scanner (Odyssey CLx, LICOR). Determination of 0.5 μg/mL DAPI staining for normalization was performed with an absorbance reader (Infinite M Plex, Tecan).

**Cell death and senescence assays**. The LDH (Pierce LDH Cytotoxicity Assay Kit; 88954, ThermoFisher Scientific) and MTS assay (CellTiter 96® Aqueous Non-Radioactive Cell Proliferation Assay; G5421 Promega) were done following the manufacturer's instructions. For LDH, conditioned medium from cells plated in a 96-well microtiter plate was analyzed by measuring absorbance at 490 and 680 nm (Infinite M Plex, Tecan). Colorimetric measurement for MTS was performed at 490 nm (Infinite M Plex, Tecan).

Senescence-associated β-galactosidase staining was determined on cells plated in six-well plates, fixed with 2% paraformaldehyde 10 min at RT and washed twice with gentle shaking 5 min at RT. Then, cells were incubated with 1 mg/mL X-gal (20 mg/mL stock in DMF; B4252, Sigma-Aldrich,) diluted in pre-warmed 5 mM potassium ferricyanide crystalline (P-8131, Sigma-Aldrich), 5 mM potassium ferricyanide trihydrate (P-3289, Sigma-Aldrich), and 2 mM magnesium chloride (M-8266, Sigma-Aldrich) in PBS at pH 6.0. Acquisition and quantification of the

images for SA-βGal and cell area was done with an automated live cell imager (Lionheart FX, BioTek).

**Comet assay.** The assay was performed according to the manufacturer's instructions (STA-351, Cell Biolabs INC.). In short, cells were plated in 6 well plates for drug treatments, collected, counted, resuspended at 100,000 cells/mL and washed with PBS at 4 °C. Cells were mixed at 1:10 with low melting agarose at 37 °C, and poured on a glass microscope slide. After cell lysis, the slides were maintained at 4 °C in alkaline buffer (pH > 13) for 20 min. After electrophoresis, the slides were washed three times with a neutralizing buffer and stained with the Vista Green DNA Dye (#235003, Cell Biolabs). All manipulations were performed protected from direct light. Analysis was performed by capturing Z-stack images with a laser confocal microscopy and measurement of Olive tail moment with the CaspLab software.

**RNA extraction and RT-qPCR.** Total RNA extraction using the TRIzol™ Reagent (15596026, Invitrogen) and cDNA synthesis using the GoScript™ Reverse Transcription Mix, Random Primers (A2800, Promega) were done according to the manufacturer's instructions. Amplification was performed with SsoAdvanced™ Universal SYBR® Green Supermix (1725271, BioRad) with 43 cycles at 95 °C for 5 s, 60 °C for 30 s and 60 °C for 1 min (for the primers see Table 2). Relative RNA expression was calculated using the comparative Ct method and normalized to the geometric mean of the GAPDH and HPRT1 mRNAs.

**Statistics and reproducibility.** Statistical analysis was performed with the aid of GraphPad Prism version 8.4 using the method specified in the legend of each figure. Exact *p*-values are specified in the figures. All quantifications were performed based on at least three independent biological replicates, sample size, number of replicates and how they are defined is specified in the figure legends. When indicated, western blots and microscopic images are shown for representative data.

**Reporting summary.** Further information on research design is available in the Nature Research Reporting Summary linked to this article.

## Data availability

The raw data for all the figures (Supplementary Data 1) and Supplementary Figures (Supplementary Material) are included as a Supplementary Data Files. All genomic sequencing data generated by the external provider (Microsynth AG, Balgach, Switzerland) for the CRISPR/Cas9 edited exon of *MAPT*, and all maps and sequences of the gene-editing plasmids and shRNA plasmids are included as supplementary material (Supplementary Data 2). All the data generated and/or analyzed, all plasmids and cell lines included in the current study are available from the corresponding authors on reasonable request.

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

## Acknowledgements

We thank the whole laboratory for support and advice during this study. We express our gratitude and thanks to the generous funding from the Synapsis Foundation, the Gelu Foundation, the AILA/OIC Foundation, the Mecri Foundation and the Swiss National Science Foundation, grant #166612. We thank our hosting institutions Neurocenter of Southern Switzerland and Ente Ospedaliero Cantonale for financial support.

## Author contributions

M.S. and C.M. designed, performed, analyzed and described most of the experiments; G.P. designed the strategy and established the knock-out cell lines; S.Pi. helped in implementing several analytical procedures; A.S. helped in creating, isolating and characterizing cell lines; S.Pa. supervised the whole study and wrote the first draft, P.P. finalized the paper; all authors revised the paper; S.Pa. and P.P. conceived and designed the study and provided the financial support for this study.

## Competing interests

The authors declare no competing financial and non-financial interests. P.P. and S.Pa. owns stocks of AC Immune SA. No funding organizations was involved in the conceptualization, design, data collection, analysis, decision to publish, preparation of the paper, or may gain or lose financially through this publication. There are no patents, products in development, or marketed products to declare.
