## [Peer Review File · Communications Biology]

Reviewers' comments:

Reviewer #1 (Remarks to the Author):

The present paper is a very provocative study that sheds light on the function of tau in a relationship with p53, which will provide an insight between the link of AD and cancer. Despite a carefully conducted research, the current manuscript has certain limitations that need to be addressed in order to make these claims:

The conclusions are based on one modified cell line, which could be an interesting effect of the non-physiological cell line.

The effect of tau could be a dual effect, since DNA protection has been reported in several studies, even in the present report. Tau phosphorylation state could be modified, changing the function and thus the outcome.

The authors investigate this effect either at transcriptional or post-translational, but they ignore the possibility of being a translational control.

In Fig 7c the levels of p53 do not agree with the western blots, or it is confusing.

In the argument that re-expression of tau restores the effect, the levels of re-expression are way above physiological.

Reviewer #2 (Remarks to the Author):

The authors investigated the role of nuclear tau and p53 in controlling chromosomal DNA damage by etoposide. In their previous report, the authors showed that Tau depletion in neuroblastoma cells severely dysregulate P53 activity and leads to reduced cell death. And, the cells are mainly senescent. The authors demonstrated here that in the absence of Tau, p53 stability, post translational modifications and activity are affected. Abnormal post-translational modifications of both P53 and MDM2 were detected in Tau-KO cells. Overall, the findings are interestingly and could be of significance in our understanding the functional mechanisms of tauopathy and neurodegeneration. However, the quality of article is poor. The authors failed to provide a mechanistic insight, and supporting evidence for Tau and p53 functional interactions and how p53 degradation are lacking. For example, 1) whether and how p53 binds Tau via which domains and/or phosphorylation need to be delineated. 2) Tau isoforms 1N3R and 1N4R are most abundant. The authors should specifically knock down both isoforms and then determine the biological effect on p53 and cell death or growth arrest. 3) Use of neuroblastoma cells cannot represent the biological effect from primary neurons. Neuroblastoma cell lines tend to be more self-protective. One cell surface protein TIAF1 in the neuroblastoma cells causes normal neurons to die when metastatic neuroblastoma cells come in close contact with neurons. 4) The authors should use wild type and mutated p53 cDNA constructs for protein overexpression and show the effect of Tau knockdown or knockout. There are more than 50% of p53 mutations among all cancer cells. One cannot guarantee that the p53 is wild type in the neuroblastoma cells used in this study. 5) The authors ignored many proteins that control the DNA damage. p53 and p53-binding partners together may affect the outcome of DNA damage. For example, p53 and WWOX are partners in apoptosis and both are crucial in maintaining chromosomal DNA integrity (Cell Cycle. 2019 Jun;18(11):1177-1186; Cell Commun Signal. 2019 Jul 17;17(1):76). In the absence of WWOX, p53 stability is decreased (J Biol Chem. 2005 Dec 30;280(52):43100-8). Also, WWOX binds Tau to prevent Tau hyperphosphorylation (J Biol Chem. 2004 Jul 16;279(29):30498-506). These should be added in the Discussion. 6) All figures are too small to read. 7) The quality of p53 antibody is no good, and this could misinterpret certain results.

Figure 1: The authors should determine the binding interactions between p53 and Tau isoforms, and

that knockdown of Tau isoforms such as 1N3R and 1N4R could provide better and convincing biological effects or significance.

Figure 2b and Supplementary Figure 2: The total % of activated caspase 3—possessing cells are only less than 15%. This is not convincing. The data should be validated using Western blotting to show the activated caspase 3 fragment. The quality of p53 is compromised. Also, knockdown effect of p53 shRNA is minimal and not convincing (Supplementary Figure 2). Two shRNA constructs should be used.

Figure 3: In addition to staining the cells with beta-galactosidase, both cell volumes and cell cycle analysis data should be shown. The extent of G1 and G2/M growth arrest should be shown.

Figure 4: The authors assume that p53 is destabilized in the Tau knockout cells. The statistics of control and the KO cells show essentially no difference (less than 1%; n= ?). The original data should be further validated and shown in the Supplemental Materials. Please show directly that p53 is destabilized in gels.

Supplementary Figure 4: Again, the p53 antibodies do not have good qualities, raising the concern of the validity of your data. pS46-p53 is involved directly in apoptosis. Antibody against pS46-p53 should be used.

Figure 5: The authors should use p53 cDNA constructs for protein overexpression and show the effect of Tau knockdown or knockout. There are more than 50% of p53 mutations among all cancer cells. One cannot guarantee that the p53 is wild type in the neuroblastoma cells used in this study.

Figure 6: I am not certain that this is the side effect of etoposide or endogenous p53 or both. It seems like etoposide does all the work. The protein levels of indicated proteins should be shown.

Dear Editor, dear Reviewers

many thanks for your time and dedication in assessing our manuscript. The comments, critics and suggestions were important in order to improve the quality of our work. Please find below a point by point rebuttal to points raised.

Because of the amount of new data requested and now delivered, additional figures were produced. For this reason, please be aware that the numbering of the figures may differ between the original submission and the current updated submission.

Reviewer #1:

The present paper is a very provocative study that sheds light on the function of tau in a relationship with p53, which will provide an insight between the link of AD and cancer. Despite a carefully conducted research, the current manuscript has certain limitations that need to be addressed in order to make these claims:

1. The conclusions are based on one modified cell line, which could be an interesting effect of the non-physiological cell line.

We have validated the effect of down-regulation of Tau on reduced P53 stabilization and apoptosis induction in two additional human neuroblastoma cell lines expressing wt P53 protein. These additional data confirmed that our observations are not limited to SH-SY5Y cells. The data are shown in the new Supplementary Figure 4 and are described in a new paragraph (Results section, page 10).

“In order to validate the observation made in SH-SY5Y cells, we tested the effect of Tau down-regulation in IMR5 and IMR32 human neuroblastoma cell lines. Similar to SH-SY5Y cells, these two cell lines express a wild-type functional P53^{26, 27}. Several other neuroblastoma cell lines were disregarded because P53 mutations were causing either constitutive activation or expression loss of P53²⁷. Tau expression in IMR5 cells was down-regulated ~4-fold in the presence of the 2112 shRNA (Supplementary Fig.4a). Under these conditions, we observed lower etoposide-induced P53 stabilization in Tau-KD when compared to mock-transduced IMR5 cells as determined by western blot and immune staining analysis (Supplementary Fig.4b). Similar to what observed in SH-SY5Y cells, etoposide-induced clCasp3 was also reduced in Tau-depleted IMR5 cells (Supplementary Fig.4c). The presence of the 3127 shRNA in IMR32 cells lowered Tau expression by

~40%, which resulted in reduced P53 stabilization and caspase-3 activation in cells exposed to the etoposide treatment (Supplementary Fig.4d-f).”

2. The effect of tau could be a dual effect, since DNA protection has been reported in several studies, even in the present report.

We share the view that Tau may be involved in multiple cellular function. A role for Tau in DNA protection has been reported previously and confirmed by us in this report. The main message of the current report is that Tau acts as a modulator of P53 stabilization and cell fate as a consequence of DNA damage. In fact, the relatively small increase in DNA damage observed in Tau-depleted cells (referred by the Reviewer) would be expected to potentiate P53 stabilization and cell death induction. Much in contrast to this, we report that P53 induction and apoptosis follow a dose-dependent increase due to etoposide exposure, Tau-depleted cells display reduced amount of P53 protein present in cells as well as reduced apoptosis induction. For this reason, we conclude that this new function of Tau is likely to be independent of its function in DNA protection, in good agreement with a dual function of Tau as suggested by the reviewer.

3. Tau phosphorylation state could be modified, changing the function and thus the outcome.

We reported previously (Ulrich et al, Sci.Rep.8:17702 (2018) that etoposide-treatment cause nuclear translocation and a reduction in the phosphorylation of overexpressed Tau. An effect of heat-stress induced DNA damage on Tau modification was reported by other groups, as mentioned in the text. It is also known that undifferentiated SH-SY5Y cells present relatively higher levels of Tau phosphorylation, similar to what observed in the embryonic brain. So, a change of phosphorylation and possibly also in the ratio of splicing variants (as suggested by Reviewer #2) may well be involved in the P53-modulating role of Tau. For the matter of the present report, the focus was on the effect of total Tau depletion rather than investigating a possible role of specific phosphorylation sites, also because to demonstrating a causal link between endogenous Tau phosphorylation and P53 modulation is a difficult task. The following paragraph was added (Discussion section, page 15).

“In the adult human brain, post-mitotic neurons express Tau in multiple alternative spliced isoforms that differ depending on the presence of up to two N-terminal inserts (0N, 1N, 2N) and on the presence of three or four microtubule-binding repeats (3R, 4R). In early development, 3R-Tau isoforms are predominant whereas in the adult brain the 3R and 4R isoforms are detected at a similar level, although they ratio is substantially altered

in a peculiar manner for distinct tauopathies⁴³. The cell lines used in our study (SH-SY5Y, IMR-32 and its subclone IMR-5) express predominantly the embryonic isoform 3R-Tau when actively dividing⁴⁴⁻⁴⁶. Our results showing that the effect of endogenous Tau deletion was reversed by ectopic expression of either 4R-Tau₄₄₁ or 3R-Tau₄₁₀ suggest that the modulatory role of Tau on P53 is possibly shared by all Tau isoforms. Tau phosphorylation in undifferentiated SH-SY5Y is increased⁴⁶, so at this time we cannot exclude that the modulatory effect of Tau on P53 may require modification of Tau by phosphorylation.”

4. The authors investigate this effect either at transcriptional or post-translational, but they ignore the possibility of being a translational control.

We do agree that the data relating to the P53 mRNA expression alone cannot exclude a translational control. Nevertheless, differential degradation rate of P53 in wt and Tau-KO cells in the presence of the translational blocker cycloheximide as well as recovery of reduced P53 in Tau-KO cells in the presence of the proteasome blocker MG132 (Fig. 7c and d) are strong argument in favor of a post-translational mechanism. Also based on a comment made by Reviewer #2, we simplified the data shown in the figure as well as the relative text aiming at facilitating its understanding (Result section, page 13).

“To address if Tau-KO cells displayed faster P53 degradation possibly accounting for the lower detection of P53 protein, we first inhibited the ubiquitin-proteasome system by treating the cells with MG132. The presence of MG132 during the recovery phase from etoposide exposure restored P53 stabilization in Tau-KO cells but had no effect in wt cells, suggesting that the absence of Tau favored P53 degradation (Fig. 7c and d, 0 h chase). Taking advantage of the fact that MG132 was able to restore similar P53 protein levels in wt and Tau-KO cells exposed to etoposide, we then analyzed the rate of degradation of P53 and MDM2 by removing MG132 and adding the translation inhibitor cycloheximide. Under these conditions we observed a faster P53 degradation rate at 2 and 4 h wash-out in Tau-KO cells when compared to wt cells (Fig. 7d). In contrast no difference was observed in terms of MDM2 degradation (Fig. 7d).”

We also modified the following sentence (Result section, page 11).

“Overall, these data essentially dismissed the premise that the effect of Tau depletion on P53 stabilization occurred at the transcriptional level, rather suggesting a translational or post-translational control.”

5. In Fig 7c the levels of p53 do not agree with the western blots, or it is confusing.

The discrepancy of the extent of the effect of Tau-depletion obtained by western blot and immune staining relates to the different nature of the technologies, although they

consistently confirmed the modulatory role of Tau on P53 stabilization. First, we may be confronted to a possible difference in the limit of detection based on a sensitivity issue, which coupled with the very low levels of P53 present at basal conditions may influence the relative amount of P53 detected in the control conditions used to determine the fold change in P53 in cells subjected to different treatments. Second, western blot analysis is performed on denatured protein resolved by electrophoretic means, whereas immune staining detects protein in their native state in cells, which because of protein conformation, modification or association with binding partners may affect the binding of antibodies and thus hamper a precise determination of the total protein amount present in cells. It is because of this assumption that we decided to perform our analysis based on both technologies. Although we share the view that it is important to precisely determine the exact extent of Tau-mediated P53 de-stabilization, performing this in an *in vitro* system that admittedly represents an oversimplified model of the *in vivo* situation is most likely of limited scientific relevance. This would not be the case when e.g. investigating a possible loss-of-function role of Tau in human tauopathies. For this purpose, we are currently developing unbiased quantitative assay for P53 quantification.

6. In the argument that re-expression of tau restores the effect, the levels of re-expression are way above physiological.

Matching ectopic expression to the level of endogenous expression is often very challenging. We provide now additional data obtained at a more physiological Tau over-expression. For this, we opted for transient transfection of a diluted Tau plasmid and analysis at three days post-transfection. The data are shown in the new Supplementary Figure 3 and are described in a new paragraph (Results section, page 9).

“Further documenting the role of Tau in etoposide-induced cytotoxicity, re-expressing high levels of human Tau₄₄₁ in Tau-KO cells (Supplementary Fig.3a) increased P53 stabilization in etoposide treated cells (Supplementary Fig.3b) and restored sensitivity in the LDH and clCasp3 assays (Supplementary Fig.3c). In order to obtain, reconstituted Tau expression at a level similar to that of endogenous Tau, in a second set of experiments Tau-KO cells were transiently transfected with a 1:10 mixture of Tau₄₁₀ and GFP plasmids or of empty and GFP plasmids. Tau expression was then analyzed in GFP-positive cells co-transfected either with the Tau₄₁₀ or the empty plasmid by immune staining. This led to determine a level of ectopic expression corresponding to ~2-fold that of endogenous Tau determined in parental SH-SY5Y cells (Supplementary Fig.3d). Under these conditions, 6 h after etoposide exposure Tau₄₁₀-transfected Tau-KO cells displayed increased P53

stabilization when compared to that detected in empty plasmid-transfected Tau-KO cells (Supplementary Fig.3e)."

Reviewer #2:

The authors investigated the role of nuclear tau and p53 in controlling chromosomal DNA damage by etoposide. In their previous report, the authors showed that Tau depletion in neuroblastoma cells severely dysregulate P53 activity and leads to reduced cell death. And, the cells are mainly senescent. The authors demonstrated here that in the absence of Tau, p53 stability, post translational modifications and activity are affected. Abnormal post-translational modifications of both P53 and MDM2 were detected in Tau-KO cells. Overall, the findings are interestingly and could be of significance in our understanding the functional mechanisms of tauopathy and neurodegeneration. However, the quality of article is poor. The authors failed to provide a mechanistic insight, and supporting evidence for Tau and p53 functional interactions and how p53 degradation are lacking.

All our data are unpublished, we do not understand to which previous report the Reviewer is referring to. The comment is misleading as it suggests that part of our work was published previously, we find it unjustified and inappropriate.

7. Whether and how p53 binds Tau via which domains and/or phosphorylation need to be delineated.

This statement is possibly made on the wrong assumption that the fact that Tau expression impacts P53 stability occurs by a direct P53/Tau interaction. We provide now evidence that we did not find a direct interaction between endogenous Tau and P53 in SH-SY5Y. As a positive control we provide evidence of the well-known P53-MDM2 interaction for endogenously expressed proteins, although this was mostly reported in an overexpressing cell model by other authors. The data are shown in the new Figure 8 and are described in a new paragraph (Results section, page 14).

"In the presence of the proteasome inhibitor, the interaction between P53 and MDM2 was confirmed by a co-precipitation experiment (Fig.8). In contrast, we did not detect any interaction between P53 and Tau (Fig.8), suggesting that the modulatory function of Tau on P53 stabilization may not occur by a direct interaction between the two proteins in SH-SY5Y cells."

8. Tau isoforms 1N3R and 1N4R are most abundant. The authors should specifically knock down both isoforms and then determine the biological effect on p53 and cell death or growth arrest.

Please kindly refer to our response to point 3 of Reviewer #1. The literature and our in-house data do not agree with the misleading statement made that “Tau isoforms 1N3R and 1N4R are most abundant”, at least for the human neuroblastoma lines used in our study but also when considering splicing in more general terms. Specifically knocking down Tau isoforms (six of them exist) is not particularly relevant, and although technical feasible, it would require a major effort without impacting the main message of the manuscript. The possible involvement of different splice variants was partly addressed in our response to point 6 of Reviewer #1.

9. Use of neuroblastoma cells cannot represent the biological effect from primary neurons. Neuroblastoma cell lines tend to be more self-protective. One cell surface protein TIAF1 in the neuroblastoma cells causes normal neurons to die when metastatic neuroblastoma cells come in close contact with neurons.

We agree with this assessment. We carefully avoided any statement that could be interpreted as that the observations made in neuroblastoma cells are the same to what may occur in normal neurons or in vivo.

10. The authors should use wild type and mutated p53 cDNA constructs for protein overexpression and show the effect of Tau knockdown or knockout. There are more than 50% of p53 mutations among all cancer cells. One cannot guarantee that the p53 is wild type in the neuroblastoma cells used in this study.

It is commonly accepted that ectopic P53 overexpression often leads to the saturation of the P53 degradation machinery, a result also obtained in our cell system (not shown). By consequence, this could mask any effect of P53 degradation rate, which is a central aspect of our work. For this reason, despite adding a substantial challenge, we deliberately focus on the biology of endogenous proteins. For the SH-SY5Y cells used in our work (as well as for IMR5 and IMR32 cells for which we have included new data), it is well established that they express wild-type P53. So, whilst we are interested to study whether Tau may modulate mutated P53, we do not think that this is critical to the current manuscript. As mentioned by the Reviewer, P53 mutations are described in >50% of cancers and they do this by several mechanisms, testing all of them is outside the scope of the current work and testing them through ectopic expression is likely to be of poor scientific relevance. The following paragraph was modified (Result section, page 10).

“In order to validate the observation made in SH-SY5Y cells, we tested the effect of Tau down-regulation in IMR5 and IMR32 human neuroblastoma cell lines. Similar to SH-SY5Y cells, these two cell lines express a wild-type functional P53^{26, 27}. Several other neuroblastoma cell lines were disregarded because P53 mutations were causing either constitutive activation or expression loss of P53²⁷.

11. The authors ignored many proteins that control the DNA damage. p53 and p53-binding partners together may affect the outcome of DNA damage. For example, p53 and WWOX are partners in apoptosis and both are crucial in maintaining chromosomal DNA integrity (Cell Cycle. 2019 Jun;18(11):1177-1186; Cell Commun Signal. 2019 Jul 17;17(1):76). In the absence of WWOX, p53 stability is decreased (J Biol Chem. 2005 Dec 30;280(52):43100-8). Also, WWOX binds Tau to prevent Tau hyperphosphorylation (J Biol Chem. 2004 Jul 16;279(29):30498-506). These should be added in the Discussion.

We acknowledge that the DDR is regulated by many factors in a complex and ample pathway, listing all of them is out of scope for a report of this kind. We made general statement such as “Not surprisingly, the P53-MDM2 axis includes a variety of factors regulating their modification and localization³⁵. Based on our data, Tau should now be listed as a modifier of wild-type P53 function, with possible implications in cancer biology.” (Discussion section, page 15) or “The balance between apoptosis and senescence is regulated by an intricate mechanism, which varies in response to distinct stressors⁴⁷. In terms of DNA damage, crucial determinants are the nature and intensity of the stress.” (Discussion section, page 15).

Nevertheless, we added the sentence (Discussion section, page 17).

“Additional P53-interacting protein such as WW domain-containing oxidoreductase (WWOX), which modulate Tau phosphorylation, may be involved in Tau-dependent regulation of P53^{53, 54}.”

12. All figures are too small to read.

This is likely to be caused by the automatic generation of the pdf file sent to the reviewers. We now provide high-definition images of all figures.

13. The quality of p53 antibody is no good, and this could misinterpret certain results.

We now increased the panel of P53 antibodies tested previously and identified two antibodies (DO-1 and Pab 1801) that based on western blot detection of endogenous P53 in lysate of cells at basal conditions, present better affinities and are more suited for this

technique. The data are further substantiated by the use of a P53-null line (see response to point 16). All P53 western blots and their quantifications were thus repeated and in part shown for both antibodies. See Fig.7c, Fig.8, Supplementary Fig.1, Supplementary Fig.2, Supplementary Fig.4, and Supplementary Fig.6. The data obtained by immune staining were not repeated because already generated with the DO-1 antibody which delivered a robust signal also at basal conditions when compared to negative control antibodies or secondary antibody alone. Nevertheless, the main message relates to DNA damage-induced P53 and we showed in the first submission and confirmed now that DNA damage leads to a strong and specific signal for endogenous P53 thereby validating the good quality of the antibodies in multiple experiments and with different technologies.

14. Figure 1: The authors should determine the binding interactions between p53 and Tau isoforms, and that knockdown of Tau isoforms such as 1N3R and 1N4R could provide better and convincing biological effects or significance.

See response to point 7

The reference to Figure 1 is wrong because this figure describes the generation of KO and KD lines and does not include any study on the effect of Tau-depletion on P53.

15. Figure 2b and Supplementary Figure 2: The total % of activated caspase 3—possessing cells are only less than 15%. This is not convincing. The data should be validated using Western blotting to show the activated caspase 3 fragment.

The low percentage of cells entering into apoptosis is due to the short and acute treatment with etoposide and the relatively short recovery time. The goal of the experimental design was indeed to allow the cells to repair the damage and thus at least in part survive the DNA damage stress. The activation of caspase 3 was confirmed by western blot (Fig.2) and the following sentence was added (Results section, page 6).

“The presence of activated cIcasp3 in Tau-expressing cells exposed to etoposide and its almost complete absence in Tau-KO cells was confirmed by western blot analysis with the same antibody for the cleaved enzyme form (Fig.2).”

16. The quality of p53 is compromised. Also, knockdown effect of p53 shRNA is minimal and not convincing (Supplementary Figure 2). Two shRNA constructs should be used.

We generated new P53-KD cells utilizing viral pseudoparticles instead of the parental plasmid, which led to significantly improved knock-down of P53. Under these conditions, etoposide-induced apoptosis was reduced by >85%. The data are shown in the new Supplementary Figure 1. The following paragraph was edited (Result section, page 8)

“To check the requirement of P53 for apoptosis induction in our cell model, we transduced cells with viral pseudoparticles and isolated stable P53 shRNA expressing cells (Supplementary Fig.1a). The effect of the shRNA was negligible at basal conditions, i.e. when the cells maintain a minimal amount of P53 due to its efficient degradation. In contrast, P53-KD cells displayed reduced etoposide-dependent P53 stabilization when compared to control cells as demonstrated by western blot analysis with the monoclonal antibodies DO-1 and Pab 1801 and confirmed by immune staining with DO-1 (Supplementary Fig.1b and c). Cell lysates obtained from the neuroblastoma cell line SK-N-AS carrying a homozygous deletion in the TP53 gene and therefore not expressing P53²⁵, were used as a negative control for P53 immune detection. Etoposide treatment induced apoptosis in ~2% P53-KD cells compared to ~14% of the control cells (Supplementary Fig.1d). These data confirmed the involvement of P53 in DSB-induced apoptosis also in SH-SY5Y cells.”

17. Figure 3: In addition to staining the cells with beta-galactosidase, both cell volumes and cell cycle analysis data should be shown. The extent of G1 and G2/M growth arrest should be shown.

The senescence-associated β Gal data were complemented by two additional analysis, p21 protein determination and mean cell size both in the knock-out and knock-down cell models. The data obtained at basal conditions and three days after etoposide treatment are consistent among the three readouts and cell models. The data are shown in Fig. 3. The following paragraph was edited (Results section, page 6).

“In alternative to cell death, unresolved DNA damage may provoke cellular senescence¹⁷. Inhibition of cyclin-dependent kinase by p21 causes cell cycle arrest and induction of senescence^{19, 20}. When compared to untreated conditions, at three recovery days after etoposide exposure, higher amounts of p21 were detected by western blot in both Tau-expressing and Tau-KO cells (Fig.3b). When comparing the extent of this effect in the absence or the presence of Tau, we found that Tau depletion increased p21 both at basal conditions as well as after etoposide treatment when compared to wt cells (Fig.3b). The increased amount of p21 present in Tau-KO cells suggests that Tau-depletion may prone cells to enter a senescence state further accelerated in the presence of DSBs. We determined the number of cells entering in a senescent state based on their mean cell size and by the senescence-associated β -galactosidase (SA- β Gal) staining procedure at mild acidic conditions. When compared to untreated conditions, significantly increased cell size and SA- β Gal-positive cells were found at three recovery days after etoposide exposure

both for wt and Tau-KO cells (Fig.3b). Again, Tau-depleted cells at basal conditions displayed a larger proportion of senescent cells in terms of cell size, SA-βGal staining and p21 expression (Fig.3b). A consistent observation was made also for Tau-KD cells when compared to control shRNA cells (Fig.3c)."

18. Figure 4: The authors assume that p53 is destabilized in the Tau knockout cells.

The statistics of control and the KO cells show essentially no difference (less than 1%; n= ?). The original data should be further validated and shown in the Supplemental Materials. Please show directly that p53 is de-stabilized in gels.

It is commonly accepted that endogenous P53 under basal conditions is present in cells at very low level due to efficient clearing, a critical feature of P53 biology. Basal P53 in Tau expressing and Tau depleted cells is not a main message of our manuscript, we report the role of Tau during the DNA damage response. All quantification of western blots and immune staining shown in Supplementary Fig.1, Supplementary Fig.2, Supplementary Fig.4, and Supplementary Fig.6 consistently demonstrate lack of evidence of an effect of Tau-depletion on P53 at basal conditions, no assumptions were made, we just report our findings.

Please note that there was no P53-related data in Figure 4 of the original submission.

19. Supplementary Figure 4: Again, the p53 antibodies do not have good qualities, raising the concern of the validity of your data. pS46-p53 is involved directly in apoptosis. Antibody against pS46-p53 should be used.

Unfortunately, we were not able to demonstrate the presence of phosphorylation at Ser46 of P53 in SH-SY5Y cells. Concerned on a possible sensitivity issue, we also tested lysates of cells obtained after prolonged, maximal etoposide induction in order to ensure optimal apoptosis induction. Also under these conditions, Ser46 phosphorylation was absent in SH-SY5Y cells. Much in contrast to Ser15 phosphorylation, which is the most characterized phospho-site of P53 in particular for its role in protecting MDM2-mediated degradation of P53, a main aspect of this report. Unbiased analysis of P53 modification by mass spectrometry will be matter of future studies. We added the following sentence (Result section, page 13).

"Our attempts to analyze pS₄₆-P53 phosphorylation was unsuccessful as no signal was detected also under conditions of prolonged etoposide treatment both in Tau-expressing and Tau-KO cells."

20. Figure 5: The authors should use p53 cDNA constructs for protein

overexpression and show the effect of Tau knockdown or knockout. There are

more than 50% of p53 mutations among all cancer cells. One cannot guarantee that the p53 is wild type in the neuroblastoma cells used in this study.

See response to point 10.

21. Figure 6: I am not certain that this is the side effect of etoposide or endogenous p53 or both. It seems like etoposide does all the work. The protein levels of indicated proteins should be shown.

The mRNA determination experiment was performed to assess the transcriptional activity of P53 on established direct gene targets modulated by P53. It is therefore not surprisingly that the most obvious effect was the robust induction of transcription when P53 is stabilized in cells exposed to a DNA damage. Our interest was on establishing whether the observed Tau-dependent modulation of P53 as a consequence of DNA damage was reflected in a difference in its P53 transcriptional activity, which indeed was reflected by different target-specific behaviors. Determining transcriptional activity at the protein levels is of poor relevance, we do not agree that this should be shown.

Nevertheless, for some targets the data are already included. For MDM2 the protein data are shown in Supplementary Fig.5 commented in the Result section, page 11: "On the other hand, etoposide treatment resulted in the expected P53-dependent upregulation of MDM2 transcription, but this was markedly reduced in Tau-KO and in Tau-KD cells (Fig.6a and b), a result that was confirmed also at the MDM2 protein level (Supplementary Fig.5)." For p21 the data are presented in Fig.3 and comment on page 11: "Interestingly, as observed for the CDKN1A protein product p21 (Fig.3c), also the CDKN1A transcript was increased in Tau-KD cells after etoposide treatment (Fig.6b), whereas this did not reach significance in Tau-KO cells (Fig.6a)." The conclusion made based on these data was improved in the following sentence: "A different regulation of direct P53-dependent genes after etoposide exposure, conveyed by a positive or negative difference in the degree of transcription activation between cells with normal or reduced Tau expression, substantiated a Tau-dependent modulation of P53 function at a post-translational level."

Reviewers' comments:

Reviewer #1 (Remarks to the Author):

The authors provided satisfactory answers.

Reviewer #2 (Remarks to the Author):

The authors have made significant revisions with the article. Tau regulation of p53 stability and function is important. The authors imply that MDM2 is involved directly in p53 stability as regulated by Tau. Overall, what is the mechanistic data regarding how Tau directly affects p53 is lacking. The relationship between p53 and MDM2 is well known. Under physiological condition, how Tau affects the p53/MDM2 complex directly or indirectly in the cytosol or the nucleus in the cell should be provided or discussed. The abstract is vague. Few concerns are:

1. "Tau acts as a modulator of P53 stabilization and cell fate as a consequence of DNA damage": This conclusion is very important. Still, the question is how p53 stabilization can be controlled by Tau.

3. New Figure 8: There is no interaction for the endogenous p53 and Tau.

I agree with your observation. Actually, under stress conditions, there are increased numbers of proteins that physically bind p53. DNA damage-inducing agents such as UV and etoposide increase the binding of p53 with many proteins. Please include in the Figure 8 with an experiment with etoposide treatment followed by co-IP. Etoposide activates JNK1 and nuclear translocation. JNK1 can hyperphosphorylate Tau, and also interacts with p53. Please refer to Genes Dev. 1998 Sep 1;12(17):2658-63.

4. Cell death:

Although we see the activation of caspase 3, do the cells die? DNA fragmentation? Propidium iodide uptake vs DAPI uptake? SubG0/1 phase of the cell cycle?

Dear Editor

many thanks for your time and dedication in assessing our manuscript. The comments, critics and suggestions were important in order to improve the quality of our work. Please find below a point by point rebuttal to points raised.

1. How p53 stabilization can be controlled by Tau?

We do lack a mechanistic explanation for our observation and this was also the main critic expressed when we submitted this manuscript to Nature Communications and Nature Cell Biology. At this point, we ask for an editorial decision, because if indeed it is required to bring forward a mechanism, this will delay disseminating our findings significantly and with these data in hands, we could go back to the journals mentioned above. We felt that it would be in the interest of the whole research community to publish our results timely, in order to have other colleagues joining the effort to understand the physiological and pathological relevance of what observed.

2. This point is missing; the reviewer goes directly from 1 to 3.

3. P53-Tau interaction after etoposide treatment?

We do recognize that, as all our data are focused on the modulatory role of Tau during the DNA damage response, we should have shown the interaction or absence of interaction in etoposide conditions, we apologize for this. Since the detection of the endogenous P53-MDM2 interaction requires inhibition of proteasomal degradation, we focused on the conditions in the presence of MG132. Fortunately, on the same gel we have also the etoposide conditions, which did not alter the outcome of the experiment and so we didn't think that these results were of particular relevance. Also following a DNA damage, we are not able to detect the P53-Tau interaction. So, we were able to modify Figure 8 by including also the third lane of the gels. In agreement with this change we added a sentence to the result section and to the legend.

4. Do cells with activated caspase 3 die?

We already provide data demonstrating that etoposide treatment cause not only caspase 3 activation but also overt cytotoxicity with the LDH and MTS assays. Moreover, in Tau depleted cells all three read-outs (LDH, MTS and caspase-3 western-blot and immunofluorescence) are reduced, another evidence of the correlation between caspase-3 activation and cytotoxicity. The requested additional assays are at this point redundant with what already provided, and we find it unfair to ask for data that were not requested during the first round of revisions, in particular if these are already shown.